# Motor memories of object dynamics are categorically organized

Evan Cesanek[1,2]*, Zhaoran Zhang[1,2], James N Ingram[1,2], Daniel M Wolpert[1,2†], J Randall Flanagan[3†]

[1]Mortimer B. Zuckerman Mind Brain Behavior Institute, Columbia University, New York, NY, United States; [2]Department of Neuroscience, Columbia University, New York, NY, United States; [3]Department of Psychology and Centre for Neuroscience Studies, Queen's University, Kingston, ON, Canada

**Abstract** The ability to predict the dynamics of objects, linking applied force to motion, underlies our capacity to perform many of the tasks we carry out on a daily basis. Thus, a fundamental question is how the dynamics of the myriad objects we interact with are organized in memory. Using a custom-built three-dimensional robotic interface that allowed us to simulate objects of varying appearance and weight, we examined how participants learned the weights of sets of objects that they repeatedly lifted. We find strong support for the novel hypothesis that motor memories of object dynamics are organized categorically, in terms of families, based on covariation in their visual and mechanical properties. A striking prediction of this hypothesis, supported by our findings and not predicted by standard associative map models, is that outlier objects with weights that deviate from the family-predicted weight will never be learned despite causing repeated lifting errors.

## Editor's evaluation

This paper provides compelling evidence from several behavioural experiments that recently learned estimates of the mass of novel objects possess a categorical structure in memory. It further links this categorical structure to important aspects of motor control, and provides a compelling window through which to consider the role of multiple systems in the learning and memory of novel object dynamics.

*For correspondence:
evan.cesanek@gmail.com

†These authors contributed equally to this work

Competing interest: The authors declare that no competing interests exist.

## Introduction

Many theories about how objects are encoded in memory have been proposed (*Collins and Quillian, 1969*; *Warrington and Taylor, 1978*; *Mervis and Rosch, 1981*; *Schacter and Cooper, 1993*; *Gauthier et al., 1999*; *Chao et al., 1999*; *Chao and Martin, 2000*; *Humphreys and Forde, 2001*; *Freedman et al., 2001*; *Ashby and Maddox, 2005*; *Kemp and Tenenbaum, 2008*; *Kriegeskorte et al., 2008*; *Kourtzi and Connor, 2011*; *Mahon and Caramazza, 2011*; *Huth et al., 2012*). These include theories concerned with the semantic, perceptual, and functional properties of objects. For example, a hammer may be semantically labeled as a tool, represented perceptually in terms of its shape, or evaluated functionally in the context of a particular task. However, the mechanical properties of objects, which are fundamentally important to human motor control, have received little attention in theories of object memory.

The majority of tasks we perform involve physical objects, and skilled interaction with these objects depends critically on our ability to predict their mechanical properties. For many of the objects that we interact with, dexterous performance requires accurate predictions of weight (*Gordon et al., 1991*; *Wolpert and Flanagan, 2001*; *Flanagan et al., 2006*; *Johansson and Flanagan, 2009*). For

example, when lifting an object from a surface, weight prediction allows us to produce the vertical forces required to raise the object smoothly. When lifting an object for the first time, people will estimate its weight based on visual information about its size and material properties (*Gordon et al., 1993*; *Flanagan and Beltzner, 2000*; *Baugh et al., 2012*; *Baugh et al., 2016*). However, once an object has been lifted, a memory is formed of its actual (i.e. directly sensed) weight, and this memory can be used to guide subsequent lifts of the object (*Baugh et al., 2012*; *Baugh et al., 2016*; *Johansson and Westling, 1988*; *Flanagan et al., 2001*; *Flanagan et al., 2008*). Thus, in addition to intact sensory and motor function, skilled manipulation—and thus the ability to perform most daily tasks—requires the capacity to form, and quickly access, representations of object weights in memory.

Here, we investigated how the mechanical properties of the myriad objects we interact with are organized in memory. To answer this question, we used a new three-dimensional robotic interface (*Figure 1a*) that, in combination with a stereoscopic virtual reality system, allowed us to simulate objects of varying size, weight, and appearance (*Figure 1b*). Objects were presented on a carousel and, on each trial, the participant 'lifted' the presented (i.e. nearest) object by first applying an upward force to the object, which was fixed to the surface of the carousel and therefore could not move. When ready, the participant pressed a button with their other hand, which caused the portion of the carousel below the object to open, releasing the object so that it was free to move. The aim was to match the upward force to the weight of the object so that it would not move up or down when released. Therefore, by measuring the force just prior to release, we could precisely measure the participant's weight prediction on every trial. Because the robot simulated the mechanics of the object, the participant received direct haptic and visual feedback about both the object's weight and their motor error (*Figure 1c*). At the end of the trial, the open portion of the carousel closed, and the participant replaced the object.

Using this task, we developed a novel motor learning paradigm in which participants repeatedly lifted a set of five similar-looking objects of varying size and weight (*Figure 1d–f*; filled circles correspond to the objects in *Figure 1b*). In our key experiment (*Figure 1d*), these objects included four training objects (the two smallest and two largest) presented in an initial training phase, and an outlier object (the middle size) introduced later in a test phase. The training objects had a common density, and therefore had a linear relationship between size and weight. Although the size of the outlier was in the middle of the training objects, its weight was greater than would be expected under the assumption that it had the same density as the training objects. Using this lifting task, we could distinguish between two high-level hypotheses about memory organization.

First, the 'object families' hypothesis asserts that multiple objects are represented in memory by clustering them into categories, or families. This hypothesis posits that the training objects and the outlier will be represented as a single family (*Figure 1d*; green line), provided that the weight of the outlier falls within the family boundary (shaded green region). As a consequence, this hypothesis predicts that participants will fail to learn the actual weight of an outlier that falls within the family boundary, and will instead estimate the weight based on the family structure (open green circle). We refer to this predicted effect as the 'family effect'. However, if the weight of the outlier is extreme and falls beyond the family boundary (*Figure 1e*), a separate memory will be formed for the outlier object. Thus, this model predicts an all-or-nothing pattern of learning whereby, depending on their family boundary, a participant will either fully learn the outlier weight or completely fail to learn it.

An alternative hypothesis is that object properties are encoded in an 'associative map'. This idea comes from a well-known theoretical framework that has been successful in explaining how sensorimotor transformations for reaching, grasping, and saccades are encoded in memory (*Zipser and Andersen, 1988*; *Salinas and Abbott, 1995*; *Pouget and Sejnowski, 1997*). In associative map models (*Figure 1d and e*; purple curve), experience with individual objects causes the visual and mechanical properties sensed during each interaction to become gradually associated. Additionally, memories of individual objects influence one another only through local generalization, producing smoothly varying mappings between visual size and expected weight. In associative map models, the predicted weight of the outlier (open purple circle) will become increasingly accurate with experience, such that an outlier of any weight will be at least partially learned. Note that associative map learning is also implemented by modular models, which have been proposed to account for learning in both motor (*Wolpert and Kawato, 1998*) and non-motor (*Kalish et al., 2004*) tasks.

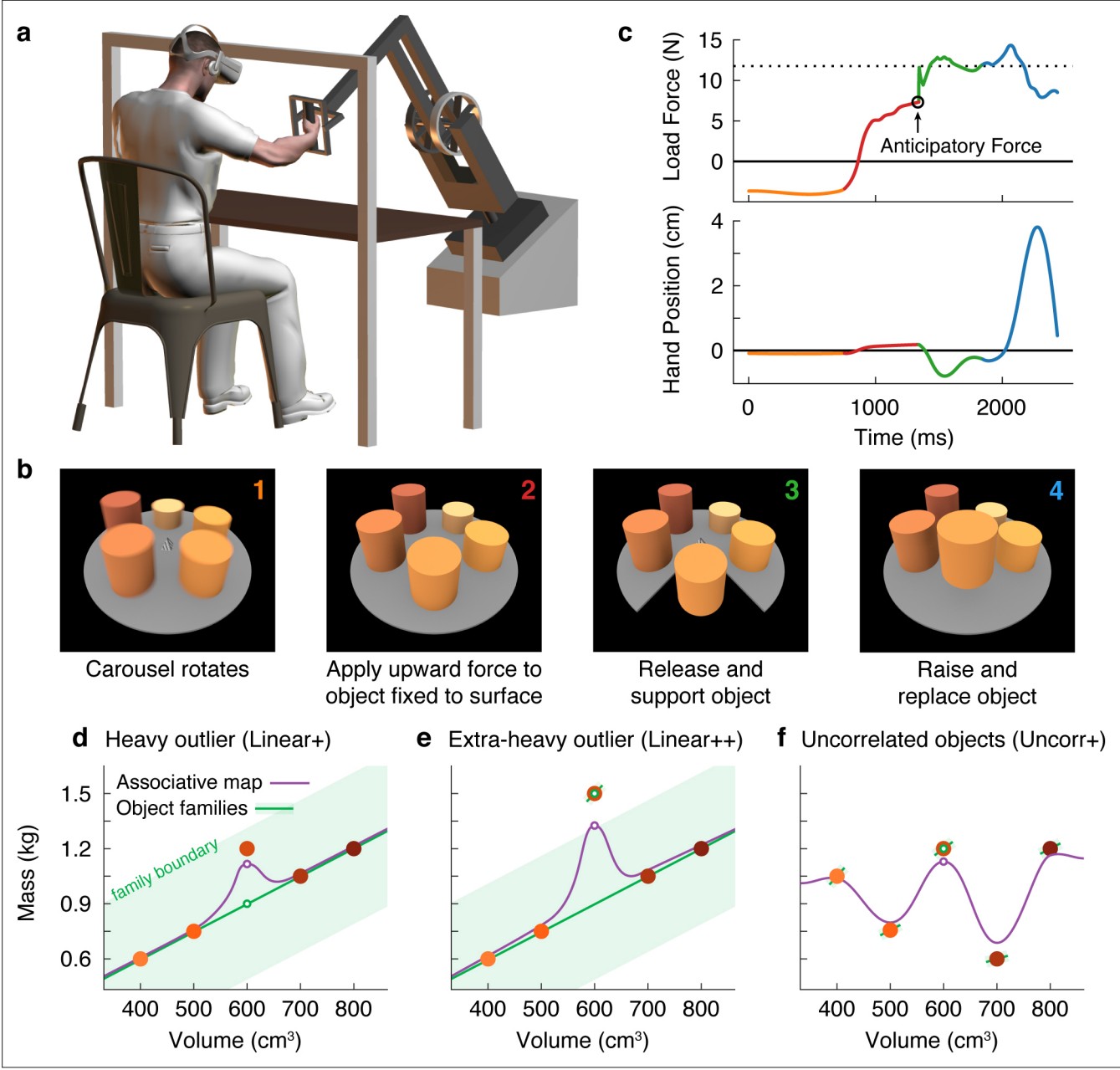

**Figure 1.** Object families and associative maps make different predictions for an outlier lifting task. (**a**) Participants grasped the handle of a three-dimensional robotic interface (3BOT) with their right hand and viewed stereoscopic scenes (Oculus Rift). The 3BOT could track movement and simulate the haptic experience of manipulating objects. (**b**) Screenshots of the key stages of the lifting task. See text for details. (**c**) Load force and vertical position traces from an example trial, color-coded to match the numbers in (**b**). In this example, the anticipatory force was less than the weight of the object (dotted line), causing a downward movement of the hand and object. (**d-f**) Tasks used to examine family representations. In these tasks there were five visually similar objects of varying volume and mass. In the Linear+ condition (**d**), four of the objects had a linear relation between size and weight. A fifth object of intermediate size had a higher density (hence the + notation) and therefore was an outlier. Under the object families hypothesis, the four objects induce learning of the family structure (green line). Visually similar objects that fall within the category boundary for the family (shaded green region) are treated as family members. Because the outlier falls within the category boundary, its weight should be persistently misestimated based on the family structure (green circle). Under the associative map hypothesis, exposure to the outlier leads to partial learning of its actual weight (purple circle). In the Linear++ condition (**e**), the object families hypothesis predicts that when the outlier becomes sufficiently extreme, and crosses the family boundary, it will be categorized as an individual and its weight fully learned. The associative map hypothesis still predicts partial learning of this outlier. In the Uncorr+ condition (**f**), when size and weight are uncorrelated, the object families hypothesis predicts that the object weights will each be learned individually. Under the associative map hypothesis, there is no fundamental difference between this scenario and those depicted in (**d, e**), so the predictions for learning are similar to the object families hypothesis (i.e. all five objects will eventually be learned).

These two hypotheses also make different predictions regarding how lifting the outlier will affect the four training objects during the test phase. Again, the object families hypothesis predicts an all-or-nothing pattern, depending on how the outlier is encoded. When encoded as a family member, the unexpectedly heavy weight of the outlier updates the family representation, causing the predicted weight to increase on a subsequent lift of a training object. However, once the outlier is classified as a separate individual, this outlier-to-family updating should be greatly suppressed. The associative map hypothesis, on the other hand, predicts that lifting the outlier will always update the estimated weights of similar-looking training objects.

Finally, the two hypotheses also make different predictions when there is no structured relationship between size and weight (*Figure 1f*). Under the object families hypothesis, each of these objects is learned as an individual (*Figure 1f*; separate green lines) and, as a consequence, the training objects will be learned more slowly than when they are learned as a family and there will be minimal single-trial generalization from the 'outlier' to the training objects. In contrast, in an associative map model, this scenario does not fundamentally differ from those depicted in *Figure 1d and e*.

Consistent with the object families hypothesis, we show that participants encode objects that covary in size and weight as a family, and that this representation exerts a powerful family effect on outlier objects, whose weights can differ markedly from the weights predicted by the family. In particular, we show that participants can completely fail to learn the weight of an outlier object, despite experiencing large, repeated movement errors; errors that, in the absence of the family, quickly drive learning. These findings address, for the first time, how motor-relevant properties of multiple objects are represented in memory.

## Results

Participants performed a lifting task in which they were required to predict the weights of five objects positioned around a carousel. *Figure 1c* shows the load force and vertical hand position in a single trial. The traces are color-coded to match the four trial phases depicted in *Figure 1b* and described above. We focused our analyses on the anticipatory force participants produced just prior to releasing the object by pressing a button with the non-lifting hand. This anticipatory force provides a precise and accurate measure of the participant's motor memory of the object weight. In the trial shown in *Figure 1c*, the participant underestimated the weight of the object, and as a consequence when the participant pressed the button to release the object, the right hand and the object moved downward. (Note that the motion of the hand after the release of the object does not provide a robust measure of participants' weight prediction because this motion depends on co-contraction and reflex responses in addition to the mismatch between vertical force and weight.)

### Motor memories of objects are organized categorically

Our initial experiment was designed to critically evaluate the object families and associative map hypotheses by examining how participants learned the weight of a heavier-than-expected outlier object. We tested separate groups of participants in the three experimental designs depicted in *Figure 1d–f*. Participants completed a training phase, in which they interacted with the four training objects, followed by a test phase, in which the fifth test object was added. All objects were visually similar—cylinders of fixed diameter with varying heights.

In the Linear+ group (*Figure 1d*), the weights of the training objects were *linearly* related to their sizes and the test object was *heavier* (as denoted by the + sign) than expected based on the training objects. The weights and sizes of the training objects ranged from 0.6–1.2 kg and 400–800 $cm^3$, respectively, and all had a density of 1.5 $g/cm^3$ (*Figure 1d*). The size of the test object was 600 $cm^3$, which was in the middle of the range of training object sizes. However, the weight of the test object, 1.2 kg, was equal to the heaviest training object, making it 0.3 kg greater than the weight that would be expected if it had the same density as the training objects.

The traces in *Figure 2a* show the anticipatory force generated for each object as a function of trial cycle (one lift of each object) across the training and test phases. The dotted horizontal lines (color-matched to the force traces) show the weights of the objects, and therefore the ideal anticipatory forces that would be generated with perfect learning. Participants in the Linear+ group very quickly learned the weights of the training objects. The scaling of forces to object weight observed in the

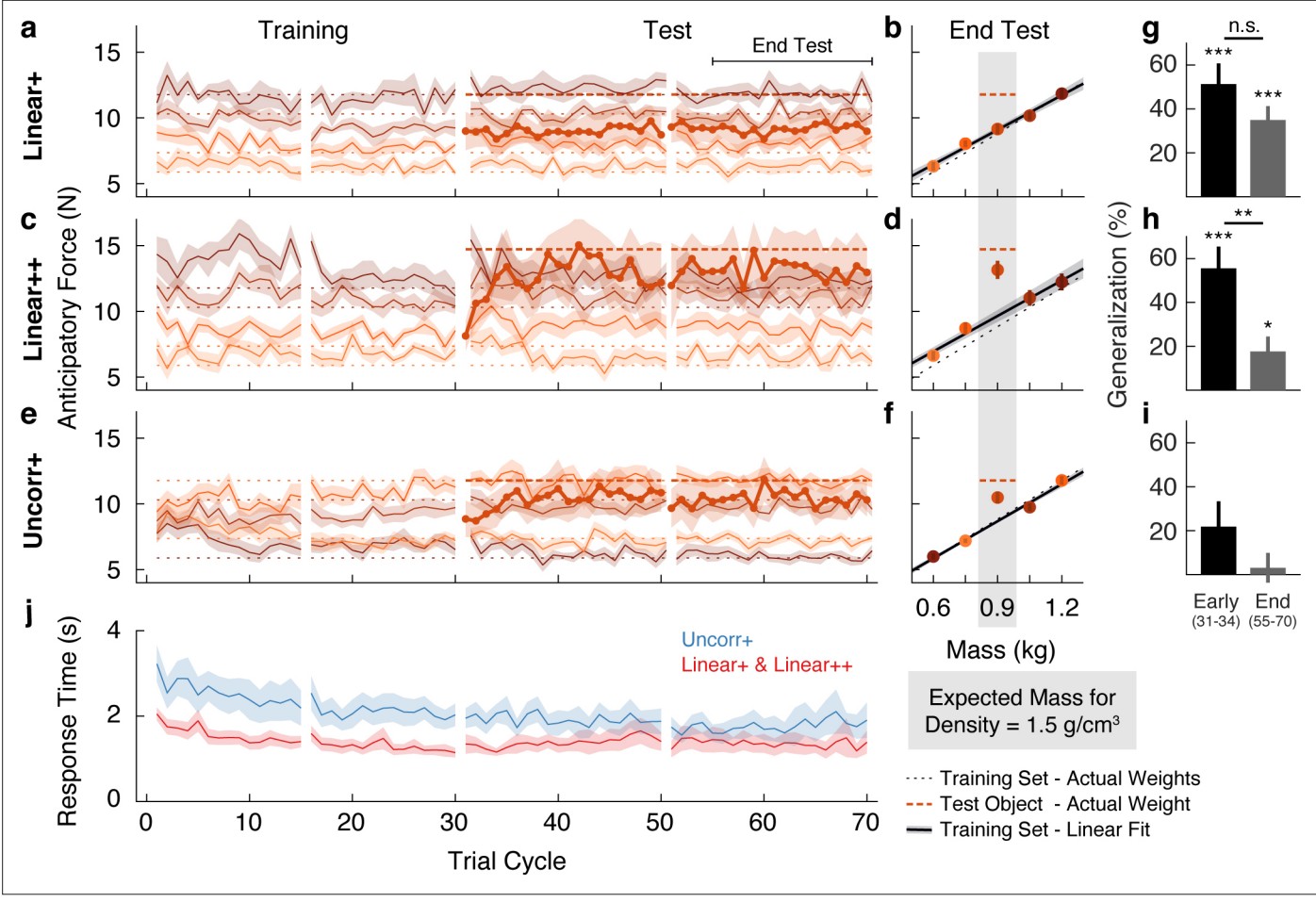

**Figure 2.** Objects are encoded according to the object families hypothesis. (**a**) Trial-by-trial anticipatory forces for the five objects over the course of the Linear+ condition (mean ± SEM). The training objects (thin lines) are experienced from the first trial cycle and the test object (thick line) is introduced on trial cycle 31 as the first trial of each cycle. Traces are color-coded with darker shades indicating larger objects and the dashed lines indicate the associated actual object weights (thick dashed line shows outlier weight). Rest breaks are indicated by gaps in the traces. (**b**) Anticipatory forces at the end of the test phase for the Linear+ condition (mean ± SEM). The abscissa shows the weights of the training objects and, for the outlier, the expected weight based on the family density. The weights of the training objects lie on the dotted unity line. Dashed horizontal line shows the weight of the outlier. Regression line shows the average of the participants' linear regressions ± SEM. (**c, d**) Same as (**a, b**) for the Linear++ condition. (**e, f**) Same as (**a, b**) for the Uncorr+ condition. Note that for each participant, the uncorrelated mapping of size and weight for the training objects was randomly selected; the shading in (**e**) and (**f**) depicts one mapping. In (**f**) the outlier is plotted at the expected weight based on the family density in the Linear conditions. (**g**) Single-trial generalization in the first four cycles (Early) and last sixteen cycles (End) of the test phase of the Linear+ condition (mean ± SEM, see Materials and methods for details). (**h, i**) Same as (**g**) for the Linear++ and Uncorr+ conditions. (**j**) Response times averaged over objects in each trial cycle (mean ± SEM). The Linear+ and Linear++ groups are combined in the red trace, as they did not differ on this measure. All SEM are across participants.

The online version of this article includes the following source code and figure supplement(s) for figure 2:

**Source code 1.** Source code for Figure 2 (Linear+, Linear++, and Uncorr+ groups).

**Source data 1.** Source data for Figure 2 (Linear+, Linear++, and Uncorr+ groups).

first trial cycle suggests that participants rapidly learned the density (a family-level parameter) based on the first few objects lifted and then used this information, in conjunction with size, to predict the weights of the other objects. At the end of the training phase (final eight cycles), anticipatory force was strongly correlated with object weight ($r = 0.76$, 95% CI = [0.66, 0.83]).

The thicker trace and dashed horizontal line, starting at trial cycle 31, show the anticipatory force and actual weight of the test object introduced in the test phase. On the first lift of the test object, the average anticipatory force was 9.00 N (95% CI = [7.68, 10.32]). This suggests that participants initially estimated that the test object would have the same density as the training objects and, therefore,

that its weight would be close to the middle of the training object weights (8.83 N). Consequently, they experienced an error of approximately 300 g (~3 N), which is close to the weight of a full can of soda and represents fully a third of the anticipated weight. Remarkably, despite this large error, participants never learned the test object weight over the 40 cycles in the test phase (40 lifts of the test object interspersed with 40 lifts of each training object). That is, the average anticipatory force did not increase—remaining at the level predicted by the family—and, therefore, participants did not adapt to the actual weight of this pronounced outlier.

We calculated the anticipatory forces at the end of the test phase (final 16 cycles) as a function of mass for the four training objects, and as a function of expected mass based on the density of the training objects for the test object (*Figure 2b*). To assess learning at the end of the test phase, we compared the average anticipatory force produced for the test object (9.15 N, 95% CI = [8.27, 10.03]) with the 'family-predicted weight' of the test object (9.09 N, 95% CI = [8.64, 9.54]), defined as the weight of the test object predicted from the best-fitting regression line through the training objects (thereby adjusting for any prediction error on the training objects). We found that the anticipatory force was not significantly greater than the family-predicted weight ($t(13) = 0.17$, $p = 0.43$).

The above results support the object families hypothesis by showing that even when the weight of an outlier object deviates markedly from its family-predicted weight, it continues to be encoded as a family member despite sensory evidence to the contrary. Next, we investigated whether there is a threshold to the family effect. We hypothesized that when the discrepancy between actual and family-predicted weight exceeds some threshold, the object will be encoded as an individual, separate from the family, despite its family-like appearance. To probe this threshold, we tested a Linear++ group, who completed the same task as the Linear+ group but with an *even heavier* outlier (hence the ++). Specifically, for the Linear++ group, the test object weighed 1.5 kg, making it 600 g heavier than if it had the same density as the training objects, and 300 g heavier than the heaviest training object (*Figure 1e*).

*Figure 2c* shows the average anticipatory force timelines for the Linear++ group. As expected, at the end of the training phase, anticipatory force was strongly correlated with object weight ($r = 0.85$, 95% CI = [0.72, 0.92]). On the first lift of the test object, participants generated an average anticipatory force of 8.13 N (95% CI = [7.19, 9.08]), consistent with the density of the training objects. However, in contrast to the Linear+ group, over the following 5–10 cycles, participants increased their anticipatory force for the test object, reaching an asymptote just below the actual object weight (14.72 N). At the end of the test phase (*Figure 2d*), the anticipatory force for the test object (13.15 N, 95% CI = [11.56, 14.74]) was significantly greater ($t(8) = 3.34$, $p = 0.0051$) than the family-predicted weight (9.65 N, 95% CI = [8.62, 10.68]).

The results of the Linear++ group demonstrate that there is a limit to how deviant an outlier object can be, with respect to a known family, before it is 'kicked out' of that family and learned as a unique individual. That is, when the error signals received from a particular object are sufficiently large, they promote the formation of a separate memory. Note that the adaptation to the test object in the Linear++ group demonstrates that participants could visually distinguish the test object from the neighboring training objects. Thus, we can conclude that the striking failure to learn the test object in the Linear+ group is not due to an inability to visually identify the test object amongst the similar-looking training objects.

Lastly, we designed a third variant of the task, in which the test object was the same size and weight as in the Linear+ group but the training objects were not related by any family structure (*Figure 1f*). Specifically, in the Uncorr+ group, the sizes and weights were remapped (separately for each participant), such that size and weight of the training objects were either completely or close to completely uncorrelated ($|r| < 0.3$). The object families hypothesis makes two key predictions for this condition. First, in the absence of structured covariation between visual and mechanical properties within the training set (i.e. when the training objects do not share a constant density), participants should be forced to form a separate memory for each training object, with no family-level representation. This, in turn, should result in slower initial learning of the training objects in comparison to the Linear groups, where all four training objects could be encoded as a family with a common density. Second, in the absence of a family representation, participants in the Uncorr+ group should be able to learn the weight of the 1.2 kg test object, unlike participants in the Linear+ group. In contrast, under the

associative map hypothesis, the results of the Uncorr+ group should not fundamentally differ from the Linear+ group.

*Figure 2e* shows the anticipatory force timelines for the Uncorr+ group. In the earliest trial cycles, there was poor differentiation of the object weights, showing that uncorrelated mappings are more difficult to learn than linear mappings. Nevertheless, by the end of the training phase the Uncorr+ group achieved accuracy comparable to the Linear groups, with anticipatory force being strongly correlated with object weight ($r$ = 0.72, 95% CI = [0.62, 0.80]). On the first lift of the test object, participants produced 8.85 N (95% CI = [7.62, 10.08]) of anticipatory lift force, which is similar to the mean of the training object weights (8.83 N). Moreover, it is similar to the force generated by participants in the Linear+ group on their first lift of the test object (9.00 N). Thus, the initial weight estimation error for the test object was similar in the Linear+ and Uncorr+ groups. However, as can be seen in *Figure 2e*, during the test phase participants in the Uncorr+ group succeeded in adapting their anticipatory force for the test object. Unlike the Linear groups, the training objects in the Uncorr+ group did not have a common density, and therefore we compared the anticipatory force for the test object to the average weight of the training objects (as the test object was of intermediate volume). At the end of the test phase (*Figure 2f*), participants' anticipatory force for the test object (10.48 N, 95% CI = [9.50, 11.46]) was significantly greater ($t(11)$ = 4.06, p = 0.00094) than the average force for the training objects (8.68 N, 95% CI = [8.44, 8.92]). The learning of the test object observed in the Uncorr+ group confirms that the failure to learn the test object in the Linear+ group is due to the structured object family, rather than the lack of a sufficient error signal.

The object families hypothesis predicts that when lifting an object that is encoded as a family member, the experienced density will update the density estimate for the family, thereby biasing the anticipatory force on a subsequent lift of a training (i.e. family) object. Conversely, when lifting a test object that is encoded as an individual, the experienced density will not update the family estimate and the anticipatory force on a subsequent lift of a training object will be unaffected. Thus, at the end of the test phase, the object families hypothesis predicts strong generalization for the 1.2 kg outlier, but no generalization for the 1.5 kg outlier. In contrast, the associative map model predicts strong generalization for the 1.2 kg outlier, and even stronger generalization for the 1.5 kg outlier. To compare these predictions, we analyzed single-trial generalization at the start and end of the test phase (*Figure 2g–i*). Specifically, we examined how the anticipatory force applied to these training objects changed when they were lifted immediately after the test object, compared to when they were lifted in the final four trial cycles of the training phase, before the test object was introduced. Note that we opted to analyze only the trials immediately following the test object because generalization is washed out with each subsequent lift of a training object, such that including these subsequent trials weakens the analysis. For the Linear+ group (*Figure 2g*), we found significant generalization both at the start ($t(13)$ = 5.47, p = 1.1e−4) and the end of the test phase ($t(13)$ = 5.56, p = 9.2e−5), with no significant change ($t(13)$ = 1.68, p = 0.12). That is, at both time points, there was an increase in anticipatory force on the trial after the test object, consistent with encoding the test object as a family member. For the Linear++ group (*Figure 2h*), there was significant generalization at the start of the test phase ($t(8)$ = 5.61, p = 5.1e−4) that was greatly reduced at the end of the test phase, with a significant change over time ($t(8)$ = 3.95, p = 0.0042). This shows that participants initially encoded the extreme outlier as a family member, but then formed a separate memory of this object. Note that although generalization was dramatically reduced in the Linear++ group, it remained significant at the end of the test phase ($t(8)$ = 2.59, p = 0.032). For the Uncorr+ group (*Figure 2i*), we found no evidence of generalization at the start ($t(11)$ = 1.90, p = 0.085) or the end of the test phase ($t(11)$ = 0.45, p = 0.66), and no change over time ($t(11)$ = 1.90, p = 0.084), consistent with encoding each object individually (*Figure 2i*).

We also analyzed the response time, defined as the time from object presentation to the button press that released the object, which is presumably linked to the time required to estimate the weight of the object. For this analysis, we combined the two Linear groups. As shown in *Figure 2j*, response times decreased during the training phase for both the linear and the uncorrelated size-weight mappings, but there was a consistent temporal cost associated with movement preparation when size and weight were uncorrelated as compared to linearly related. To assess these effects, we defined four epochs by splitting both the training and test phases into two equal parts. A two-way repeated-measures ANOVA on log-transformed response times revealed significant main effects of Group ($F$(1,

33) = 5.79, p = 0.022) and Epoch (*F*(3, 99) = 13.039, p = 0.30e−7), but no interaction (*F*(3, 99) = 0.80, p = 0.49). Separate *t*-tests on each epoch all showed significant Group effects (p < 0.048 in all four epochs). These results show that, even at the end of the test phase, encoding each object individually resulted in a temporal cost compared to encoding the objects as a family.

The increased response time in the Uncorr+ condition could be a 'switch cost' related to loading a different category in motor memory. If there is a switch cost then we should observe an increase in response time in the Linear++ condition when lifting the outlier after lifting a family member, and when lifting a family member after the outlier, as both of these situations involve switching between categories. However, we found no evidence of increased response times on these 'switch trials'. We first determined the predicted response times assuming there is no switch cost. These predictions were based on the response times observed when lifting a family member after having lifted a family member in the previous trial, that is, where the category did not change (see Materials and methods). In the Linear++ condition, the average response time when lifting the outlier after a family member group was only 3 ms longer (95% CI = [−156, 159]) than the predicted value, and the average response time when lifting a family member after the outlier was 25 ms faster (95% CI = [−129, 79]) than the predicted value. Thus, our data provide no evidence for the idea that the large temporal cost observed in the Uncorr+ condition is a switch cost. However, our data are consistent with the idea that there is a 'selection cost' associated with mapping the visual stimulus onto the appropriate category. In the Uncorr+ group, participants were required to map each visual stimulus onto one of five categories in memory, and thus we would expect a substantial selection cost. In contrast, in the Linear++ condition, only two categories were involved, and therefore the selection cost may be negligible.

### Re-organization of motor memories of objects

In the experiment described above, for the Linear groups we first introduced a set of objects with a common density, before adding in a test object, or outlier, with a higher density. We found a strong family effect such that participants never learned the weight of a test object that was 300 g heavier than expected. A key question is whether exposure to an object family can lead to the reorganization

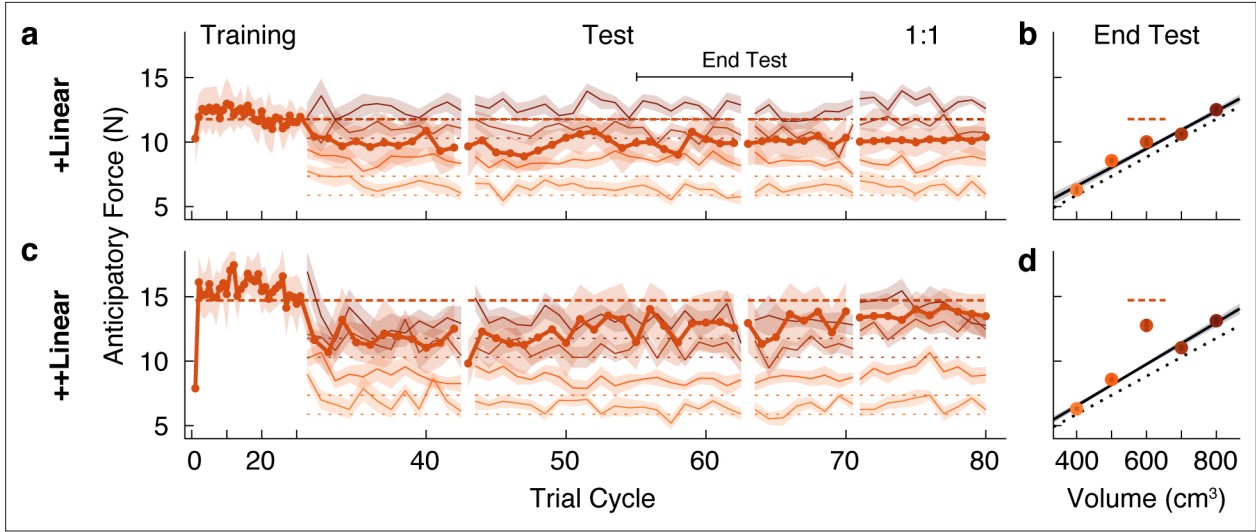

**Figure 3.** A memory of an individual is reorganized when an object family is introduced. (**a, c**) Trial-by-trial anticipatory forces, as in *Figure 2a and c*, in a 'reverse' condition in which the outlier object was learned during the initial training phase, and the family objects were only introduced from trial cycle 31. Hence, we refer to these as +Linear and ++Linear. As the training phase trial cycles contained only one trial (the outlier), for clarity, the abscissa scale is compressed. After the test phase, in a '1:1' phase the test object was presented four times in each trial cycle (rather than once as in the test phase), with each family member presented once (eight trials per cycle) such that the participant experienced the test object as often as a family member. For the 1:1 phase, we excluded trials from analysis in which the outlier object followed itself. (**b, d**) Average anticipatory forces at the end of the test phase, as in *Figure 2b and d*, but here plotted by volume.

The online version of this article includes the following source code and figure supplement(s) for figure 3:

**Source code 1.** Source code for Figure 3 (+Linear and ++Linear groups).

**Source data 1.** Source data for Figure 3 (+Linear and ++Linear groups).

of an existing memory of an individual object. To address this question, we tested two new groups of participants on conditions in which the test object was experienced *before* the four common-density 'family' objects. Note that we used the same family and test objects as in our first experiment. We refer to these groups as the +Linear and ++Linear groups to denote the reversed order in which participants encountered the test object and the family objects. In the initial training phase, participants in the +Linear group lifted the 1.2 kg test object, and the ++Linear group lifted the 1.5 kg test object. For both groups, the four family objects were then introduced in the test phase.

As expected, both groups quickly and accurately learned the weight of the test object when it was presented individually during the training phase (*Figure 3a and c*). However, at the start of the test phase (beginning at trial cycle 31), it is evident that participants in both groups began to treat the outlier and the four family objects as a single family. Specifically, the estimated weight of the test object (i.e. the anticipatory force) decreased towards the family-predicted weight. At the same time, the estimated weights of the family members were initially overestimated. These results show that even brief exposure to an object family can reorganize the memory of a previously learned individual object, such that it is assimilated into the family.

Following this assimilation of the test object, or outlier, into the family, the pattern of results is strikingly similar to that observed in our first experiment. Specifically, participants in the +Linear group never fully re-learned the actual weight of the outlier, whereas participants in the ++Linear group adapted their anticipatory force to the actual weight. At the end of the test phase (*Figure 3b*), the anticipatory force for the outlier in the +Linear group (10.01 N, 95% CI = [9.18, 10.84]) was not significantly greater ($t(10) = 1.23$, $p = 0.12$) than the family-predicted weight (9.49 N, 95% CI = [9.11, 9.86]). Thus, participants in the +Linear group did not re-learn the actual weight of the outlier after it was assimilated into the family. Therefore, the +Linear group, like the Linear+ group, exhibited a strong family effect. In the ++Linear group, the anticipatory force for the outlier at the end of the test phase (12.76 N, 95% CI = [11.14, 14.39]) was significantly greater ($t(10) = 4.19$, $p = 0.00093$) than the family-predicted weight (9.76 N, 95% CI = [9.47, 10.04]). Thus, as was the case for the Linear++ group, the ++Linear group exhibited learning (or re-learning) of the more extreme outlier.

The failure to learn the weight of the outlier in the Linear+ and +Linear groups could be due to the fact that the higher density outlier was lifted only once for every four lifts of the family objects. Thus, after the test phase we included a '1:1' phase where the relative frequency with which the outlier and family objects were experienced was equivalent. Specifically, this phase consisted of ten cycles in which the outlier object was lifted four times per cycle and each family member was lifted only once, for a total of eight lifts per cycle with the outlier and family members randomly interleaved. As shown in *Figure 3a*, in the +Linear group there was minimal impact on learning in the 1:1 phase. In the ++Linear group, increasing the relative frequency of outlier lifts in the 1:1 phase did not further improve the separation between the anticipatory force for the outlier and its family-predicted weight. These findings demonstrate that the family effect cannot be accounted for by the greater relative frequency of the family objects.

## Category boundaries are flexible

In the first two experiments, we showed that participants failed to learn the weight of a test object, or outlier, that was 300 g (or 33%) greater than the weight predicted by the density of the family, but did learn the weight when the test object exceeded this weight by 600 g (or 67%). This suggests that there is a boundary, between these two weights, that determines whether the object is encoded as a family member or as a separate individual. A fundamental question is whether such boundaries are fixed or flexible. Research on both perceptual and conceptual categorization has shown that category boundaries may depend on within-category variability (*Rips, 1989*; *Huttenlocher et al., 2000*; *Clayards et al., 2008*), and that category labeling can exhibit hysteresis whereby the point at which the perceived category changes depends on the direction of change (*Williams et al., 1986*; *Hock et al., 1993*; *Poltoratski and Tong, 2014*). To examine this issue in relation to object categorization, we recruited two new groups of participants who initially experienced the same conditions as the Linear+ and Linear++ groups from our first experiment. That is, both groups completed a training phase in which they lifted the four family objects, followed by a test phase in which the test object was initially either 1.2 or 1.5 kg for 20 trial cycles. However, we then gradually changed the test object's weight by steps of 50 g every eight trial cycles. In the Linear↗ group, the weight was gradually

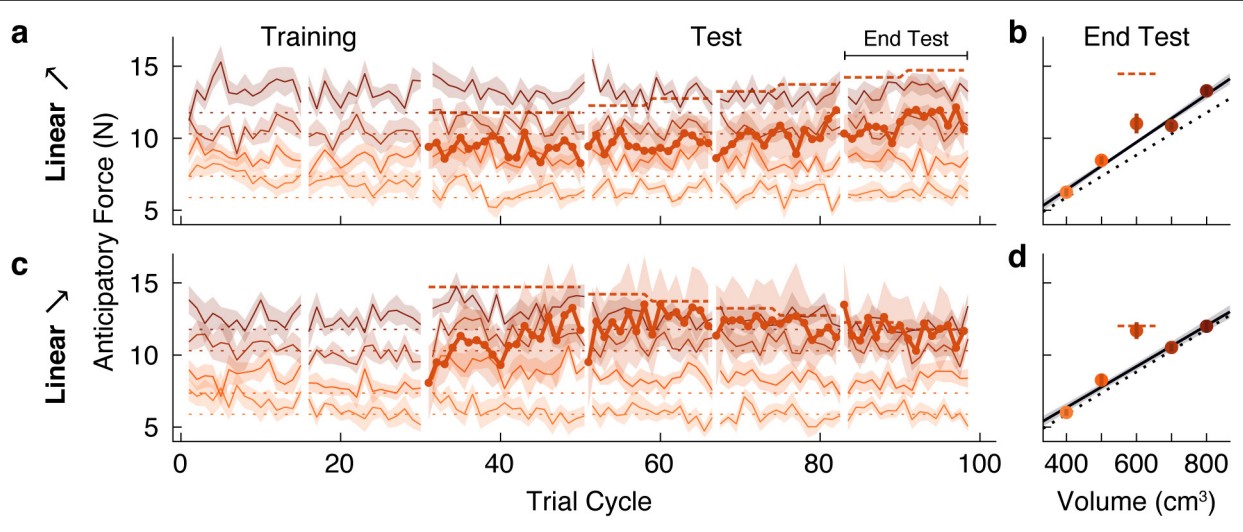

**Figure 4.** Family boundary depends on history of sensorimotor experience. (**a**) Trial-by-trial anticipatory forces (same format as *Figure 2a*) in an 'increasing' condition (Linear↗) in which the outlier starts at the weight of the Linear+ group on trial cycle 31 and increases gradually to the weight of the Linear++ condition. (**b**) Anticipatory forces at the end of the test phase (same format as *Figure 2b*). (**c,d**) Same as (**a,b**) for a 'decreasing' condition (Linear↘) in which the outlier starts at the weight of the Linear++ condition and decreases gradually to the weight of the Linear+ condition.

The online version of this article includes the following source code and figure supplement(s) for figure 4:

**Source code 1.** Source code for Figure 4 (Linear↗ and Linear↘ groups).

**Source data 1.** Source data for Figure 4 (Linear↗ and Linear↘ groups).

increased from 1.2 to 1.5 kg and in the Linear↘ group, the weight was gradually decreased from 1.5 to 1.2 kg.

The anticipatory force data for the Linear↗ and Linear↘ groups (*Figure 4a and c*) contain several features that replicate the key findings from our first experiment. First, both groups quickly and accurately learned the weights of the training objects, with anticipatory forces that were strongly correlated with actual object weights by the end of the training phase ($r = 0.81$, 95% CI = [0.73, 0.87] in Linear↗; $r = 0.84$, 95% CI = [0.77, 0.89] in Linear↘). Second, in both groups the anticipatory force generated on the first lift of the test object was close to the middle of the weights of the family objects (9.40 N, 95% CI = [8.10 10.70] in Linear↗; 8.07 N, 95% CI = [5.93, 10.22] in Linear↘). Third, at the end of the initial 20 cycles of the test phase, during which the test object weight remained at its initial value, learning of the 1.2 kg test object was not significant (Linear↗: $t(8) = -0.58$, $p = 0.71$), whereas learning of the 1.5 kg test object was significant (Linear↘: $t(8) = 2.15$, $p = 0.032$).

For the Linear↗ group, the anticipatory force for the test object does appear to have slightly increased as its weight increased. However, the anticipatory force at the end of the test phase (11.02 N, 95% CI = [9.44, 12.60]) was not significantly greater ($t(8) = 1.81$, $p = 0.054$) than the family-predicted weight (9.72 N, 95% CI = [9.43 10.01]), and was still substantially less than the actual weight (14.72 N; *Figure 4b*). Thus, despite the fact that the test object weighed 1.5 kg at the end of the test phase, it was not 'kicked out' of the family, in contrast to the equally heavy test object experienced by the Linear++ group in our first experiment. A direct comparison between the Linear↗ and Linear++ groups showed a significant difference in the anticipatory force for the outlier object at the end of the test phase ($t(16) = 2.20$, $p = 0.043$).

As noted above, and as expected based on the Linear++ group, participants in the Linear↘ group increased their anticipatory force for the 1.5 kg test object from the start of the test phase, before its weight began decreasing. Then, as the anticipatory force increased and the actual weight of the test object gradually decreased, these two forces became closely matched, and remained so until the end of the test phase (*Figure 4c*). At the end of the test phase, the anticipatory force (11.69 N, 95% CI = [10.35, 13.03]) was significantly greater ($t(8) = 4.35$, $p = 0.0012$) than the family-predicted weight (9.20 N, 95% CI = [8.65, 9.75]) and indistinguishable from the actual weight (11.77 N; *Figure 4d*). Thus, once a separate memory was formed for the test object, it continued to be encoded as an individual

even when its weight deviation decreased to the level ( + 300 g, or 33%) that the Linear+ group failed to learn. A direct comparison between the Linear↘ and Linear+ groups showed a significant difference in the anticipatory force for the outlier object at the end of the test phase ($t(21) = −3.68$, p = 0.0014). Overall, the results from both groups demonstrate that the threshold for categorizing an object as either a family member or an individual object is flexible and depends on past sensorimotor experience. Mechanisms that could potentially give rise to these effects are discussed below.

## All-or-nothing learning of outlier weight

According to the object families hypothesis, an outlier object is encoded categorically as either a family member or an individual. As a consequence, a given participant should either fully learn the weight of an outlier object or not learn at all, depending on their particular threshold for 'kicking out' an object from a family. Assuming that the threshold weight at which an outlier is kicked out of a family varies across participants, the object families hypothesis predicts that for certain outliers, there will be a bimodal distribution of estimated weights across participants (separating learners from non-learners). In contrast, the associative map hypothesis predicts that partial learning will be observed and that, assuming learning rates across participants are normally distributed, there will be a unimodal distribution in the amount of learning, regardless of the weight of the outlier.

With the aim of examining distributions across participants, we performed a web-based experiment in which we recruited a large number of participants (N = 196), divided into four groups that varied in how the outlier deviated from a linear family. As in our first experiment, we tested groups who were presented with an outlier object that was heavier (Linear+) or much heavier (Linear++) than the weight predicted by the density of the training objects. In addition, to assess the generality of our findings, we tested groups who were presented with an outlier that was lighter (Linear-) or much lighter (Linear--) than the weight predicted by the density of the training objects.

Based on the object families hypothesis, we expected that the participants in the groups with less deviant outliers (Linear+ and Linear-) would form a single distribution of non-learners, with anticipatory forces centered on the family-predicted weight. In contrast, we predicted that participants in the more deviant outlier groups (Linear++ and Linear--) would cluster into distinct distributions of learners and non-learners, with anticipatory forces centered on the actual and family-predicted weights of the outlier, respectively.

The web-based task was designed to closely mirror the laboratory task. The visual scene consisted of five cylindrical objects each with a spring attached to its top (*Figure 5a*). The objects were clamped in place by a ring that rotated before each trial to bring one of the objects to the foremost position. Participants used their mouse or trackpad to stretch the spring upwards in an attempt to generate a lifting force on the object that matched its weight (trial phase 1). Then, they pressed a key with their other hand to release the clamp (trial phase 2). From this point on, the object's motion was simulated as a mass-spring-damper system, thus providing visual feedback about the participant's performance. If the spring was stretched too much (or too little), the object would rise (or fall) and then oscillate until coming to rest (*Figure 5a*, rightmost panel). The oscillation time depended on the mismatch between the estimated and actual object weight, creating a natural time penalty. Note that although the exact nature of the sensory information in the web-based task differs from the laboratory task, both are fundamentally motor control tasks as they test how individuals translate sensory information into continuous motor commands to achieve an action goal.

The results for the Linear+ and Linear++ groups in the web-based experiment (*Figure 5b and e*) were very similar to those observed for the corresponding groups in our first experiment. This indicates that similar learning processes were engaged despite the use of visual dynamics without haptic feedback (*Danion et al., 2012*). On average, the Linear+ group did not learn the outlier, whereas the Linear++ group exhibited substantial, but not complete, learning. Our analysis, however, focused on the distributions of anticipatory forces for the outlier object at the end of the test phase (final five cycles) across participants in each group (*Figure 5c and f*). For each distribution, we fit a single-Gaussian and a two-Gaussian mixture model (blue and green curves, respectively). To compare these models, we computed the difference in the Akaike Information Criteria (ΔAIC), with positive values in favor of the two-Gaussian mixture, and we report the relative likelihood for the favored model. As expected, for the Linear+ group, in which learning of the weight of the outlier was not observed, the single-Gaussian model was favored (ΔAIC = −4.6; relative likelihood = 10.0). In contrast, for the

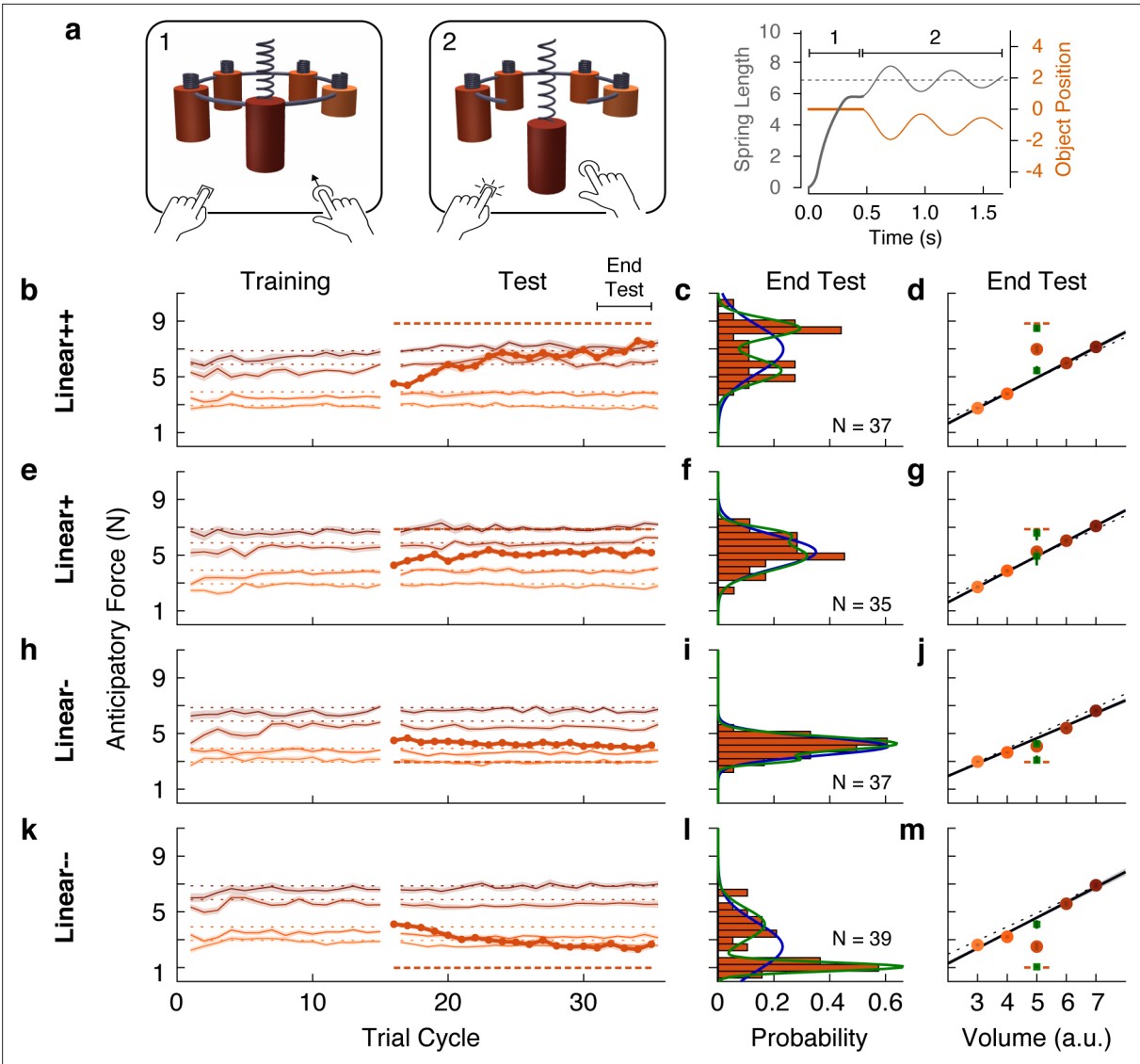

**Figure 5.** Individual differences show that outliers are either fully learned or not learned at all. (**a**) Web-based lifting experiment. (1) Five visually similar objects were clamped onto a ring, which rotated to bring the target object to the front. Participants clicked and dragged upward using their mouse or trackpad to stretch a spring, thereby applying a lifting force to the object. (2) When ready, they pressed a key on the keyboard with their other hand to release the object from the ring. The object and spring were simulated as a mass-spring-damper providing visual feedback about performance, with greater errors giving rise to larger oscillations, which also took longer to decay. As in the laboratory experiments, the goal was to prevent the object from moving after the key press. Right column shows the spring length (i.e. lift force, gray) and object position (orange) traces for an example trial in which the anticipatory force was less than the object weight. (**b, e, h, k**) Trial-by-trial anticipatory forces (formatted as in *Figure 2a*) for four conditions: two with a heavy outlier (Linear+ and Linear++, as in *Figure 2*) and the others with a lighter (Linear-) or much lighter (Linear--) outlier. (**c, f, i, l**) Histograms show the distribution across participants of the average anticipatory force for the outlier object at the end of the test phase. Blue and green curves show the fits of a single-Gaussian and a two-Gaussian mixture model, respectively. (**d, g, j, m**) Anticipatory forces at the end of the test phase (as in *Figure 2b*). The mean of each Gaussian component of the two-Gaussian mixture model is plotted as a green square, with standard error estimated via parametric bootstrap.

The online version of this article includes the following source code and figure supplement(s) for figure 5:

**Source code 1.** Source code for Figure 5 (web-based experiment).

**Source data 1.** Source data for Figure 5 (web-based experiment).

Linear++ group, the distribution was clearly bimodal, separating participants who either did or did not learn the outlier weight. This bimodal distribution was better captured by the two-Gaussian model (ΔAIC = 7.0, relative likelihood = 33.1).

For the Linear+ and Linear++ groups, the average anticipatory forces applied to the five objects at the end of the test phase are shown by the filled circles in *Figure 5d and g*. The mean of each Gaussian component of the two-Gaussian mixture is shown as a green square. In the Linear++ group, the greater of these two means (8.48 N, 95% CI = [7.98, 8.89])—representing the learners—lies almost perfectly on the actual outlier weight (dashed line, 8.83 N), whereas the lesser of the two means (5.43 N, 95% CI = [4.91, 6.08])—representing the non-learners—is very close to the family-predicted weight (4.91 N, 95% CI = [4.79, 5.03]). Surprisingly, although the single-Gaussian model was favored for the Linear+ group, one can nevertheless see two peaks in the two-Gaussian model (6.59 N, 95% CI = [5.58, 7.18] and 4.93 N, 95% CI = [3.66, 5.28]) that, respectively, closely match the actual weight (6.87 N) and family-predicted weight (4.93 N, 95% CI = [4.72, 5.13]) of the outlier. Thus, while most participants in the Linear+ group did not learn the outlier weight at all, there was a small subgroup who fully learned this weight.

The same pattern of results was observed for the Linear- and Linear-- groups (*Figure 5i–j and l–m*). For the Linear- group, the distribution of anticipatory forces for the outlier object at the end of the test phase were best fit by the single-Gaussian model (ΔAIC = −3.7, relative likelihood = 6.4), whereas the two-Gaussian model was preferred for the Linear-- group (ΔAIC = 29.3, relative likelihood = 2.3e + 6). For the Linear-- group, the means of the two components of the two-Gaussian model (1.05 N, 95% CI = [0.91, 1.22] and 4.10 N, 95% CI = [3.47, 4.68]) were, respectively, very close to the actual weight (0.98 N) and family-predicted weight (4.58 N, 95% CI = [4.36 4.79]) of the outlier. As was the case for the Linear+ group, the two-Gaussian mixture model fit to the Linear- group picked out a cluster of non-learners and a smaller cluster of learners, whose means (3.08 N, 95% CI = [2.75, 3.77] and 4.26 N, 95% CI = [4.07, 4.73]) respectively correspond to the actual weight (2.94 N) and family-predicted weight (4.66 N, 95% CI = [4.52, 4.79]) of the outlier. Overall, the results of this large-sample web-based experiment clearly support the object families hypothesis over the associative map hypothesis. At the level of single participants, the outlier was either encoded as a family member, in which case lift errors were ignored, or it was identified as a distinct individual, in which case lift errors drove complete learning of the outlier's weight.

Notably, similar bimodality was also observed in the laboratory experiments. Revisiting these data, we applied the same mixture model analysis to individual participants' final outlier learning (i.e. the difference between the anticipatory force produced for the outlier and the family-predicted weight of the outlier). To obtain sufficient sample sizes to fit the models, we combined participants from the Linear+ and +Linear groups, and from the Linear++ and ++Linear groups. The two-Gaussian model outperformed the single-Gaussian model for the combined Linear+ and +Linear group (ΔAIC = 13.3, relative likelihood = 772.8) and the resulting clustering yielded 20 non-learners and five learners. In the combined Linear++ and ++Linear group, the two-Gaussian model outperformed the single-Gaussian model (ΔAIC = 9.8, relative likelihood = 134.3), with the resulting clustering yielding six non-learners and 14 learners. When considering only the non-learners in the combined Linear+ and +Linear group, the anticipatory force produced for the outlier at the end of the test phase (8.98 N, 95% CI = [8.50, 9.47]) was very similar to the family-predicted weight (9.33 N, 95% CI = [8.97, 9.68]), as expected. Strikingly, when considering the small subset of learners in the combined Linear+ and +Linear group, the anticipatory force for the outlier at the end of the test phase was 11.71 N (95% CI = [11.34, 12.08]), showing that these participants fully learned the actual weight of the outlier (11.77 N). The same pattern of all-or-nothing learning in the two clusters was found for the combined Linear++ and ++Linear group. Specifically, the anticipatory force for the outlier produced by non-learners (10.17 N, 95% CI = [8.94, 11.40]) was very similar to the family-predicted weight (10.52 N, 95% CI = [9.50, 11.54]). Note that the family-predicted weight is slightly increased for the non-learners in these conditions. This is because the family shifted to partially accommodate the extreme weight of the outlier, but it was not kicked out of the family. The anticipatory force produced by the learners in the combined Linear++ and ++Linear group (14.12 N, 95% CI = [13.39, 14.86]) was very similar to the actual weight of the outlier (14.72 N). Thus, these results strengthen our main conclusion that memories of the motor properties of objects are organized categorically.

## Discussion

We have examined how the mechanical properties of objects we interact with are represented in memory. In a series of experiments, we provide evidence that 'motor memories' of objects are organized in terms of families. More specifically, we show that when encountering a set of new objects whose size and weight covary, participants have a strong propensity to encode the objects as a family. The consequence of this encoding is that an object that appears to be part of a previously learned family, but is an outlier in terms of weight, may nevertheless be classified as a family member. In this case, participants predict the outlier's weight based on the family and never learn its actual weight. This 'family effect' on the outlier can be anterograde, such that the family interferes with learning the weight of a newly introduced outlier, or retrograde, such that an already-learned outlier weight will be forgotten when the family is introduced. We also show that there is a weight threshold at which a sufficiently deviant outlier will 'escape' the family and be learned as an individual object. Moreover, we show that the error experienced when lifting an outlier that is encoded as a family member updates the estimated weights of the other family members. However, if the outlier has been learned as an individual, such updating is not observed. Additionally, we show that the threshold that determines whether an outlier is classified as an individual or a family member depends on recent sensorimotor experience.

Two broad approaches have been used in motor control to examine how dynamics, experienced during arm and hand movements, are represented in memory. The first approach involves applying novel dynamics, or 'force fields', to the hand. Typically this has been done by asking participants to move a handle, which is attached to a robotic manipulandum and visually represented as a cursor, between visual targets located in a horizontal plane. This work has focused on the reference frame in which individual force fields are represented (*Shadmehr and Mussa-Ivaldi, 1994*; *Krakauer et al., 2000*; *Malfait et al., 2002*; *Davidson et al., 2005*; *Berniker et al., 2014*), and on contextual factors that enable people to learn two different force fields that apply forces in opposite directions (*Davidson et al., 2005*; *Brashers-Krug et al., 1996*; *Gandolfo et al., 1996*; *Krakauer et al., 1999*; *Karniel and Mussa-Ivaldi, 2002*; *Tong et al., 2002*; *Caithness et al., 2004*; *Osu et al., 2004*; *Nozaki et al., 2006*; *Howard et al., 2008*; *Addou et al., 2011*; *Howard et al., 2012*; *Howard et al., 2013*; *Sheahan et al., 2016*; *Heald et al., 2018*; *McGarity-Shipley et al., 2020*). Although force fields may, arguably, be viewed as objects (at least in some contexts; *Cothros et al., 2006*; *Cothros et al., 2009*; *Kluzik et al., 2008*), this previous work has not examined how memories of multiple objects might be organized. The second approach to investigating how dynamics are represented in memory focuses on weight prediction when lifting objects, which is critical for dexterous manipulation. This work has shown that people can exploit learned associations, or 'priors', between size and weight, and between material and weight, to estimate the weight of an object (*Gordon et al., 1991*; *Gordon et al., 1993*; *Baugh et al., 2012*; *Cole, 2008*; *Buckingham et al., 2009*). Although such priors are often useful, for many objects that we interact with they do not provide accurate weight predictions. Importantly, once an object has been lifted, people can form a long-lasting 'object-specific' memory of the object's actual weight (*Gordon et al., 1991*; *Gordon et al., 1993*; *Baugh et al., 2012*; *Johansson and Westling, 1988*; *Flanagan et al., 2001*; *Flanagan et al., 2008*). However, the question of how motor memories of the myriad objects we interact with are represented and organized has not been addressed.

Where in the brain might motor memories of objects be stored? According to a well-known neuroanatomical framework for understanding visual processing in the primate brain, the dorsal visual pathway, in parietofrontal cortex, supports visual processing for action, whereas the ventral visual pathway, in ventrotemporal cortex, supports visual processing for perception (*Goodale and Milner, 1992*). This framework arose primarily from studies examining reaching and grasping movements directed towards objects, where the relevant object properties (e.g. size, shape, location) can be directly appreciated through vision. The control of these actions involves mapping these visual features onto motor commands to move and shape the hand (*Salinas and Abbott, 1995*; *Arbib, 1981*; *Jeannerod, 1981*; *Pouget and Snyder, 2000*), and there is abundant evidence that parietofrontal cortex is engaged in such computations (*Jeannerod et al., 1995*; *Rizzolatti and Luppino, 2001*; *Castiello and Begliomini, 2008*; *Grafton, 2010*; *Battaglia-Mayer and Caminiti, 2018*). However, as emphasized above, skilled object manipulation requires knowledge of mechanical properties, which cannot be directly appreciated through vision and must instead be estimated based on object memories linking visual and mechanical properties. Some evidence suggests that such memories could involve parietal

and premotor regions of the dorsal pathway (*Chouinard et al., 2005*; *Jenmalm et al., 2006*; *Chouinard et al., 2009*; *Freedman and Assad, 2009*; *van Nuenen et al., 2012*). However, the maintenance of durable memory representations of objects is more commonly associated with the ventral visual pathway (*Bruce et al., 1981*; *Ungerleider and Haxby, 1994*; *Riesenhuber and Poggio, 1999*; *Grill-Spector et al., 2001*; *Erez et al., 2016*). Given that category selectivity is a well-established organizational feature of ventrotemporal cortex (*Kriegeskorte et al., 2008*; *Grill-Spector and Weiner, 2014*), it seems plausible that the ventral pathway also plays a role in categorizing the mechanical properties of objects. Consistent with this view, it has been shown that, in the context of lifting, object weight is represented in the lateral occipital complex (LOC) (*Gallivan et al., 2014*), an object-selective ventral region also known to be active during reaching and grasping (*Culham et al., 2003*; *Monaco et al., 2014*). On the other hand, LOC does not appear to represent object mass that can be inferred when simply viewing objects interacting (*Schwettmann et al., 2019*).

Beyond the dorsal and ventral visual pathways, several other candidate brain regions may be involved in learning object families in the service of dexterous manipulation. For instance, predictive encoding of object weight has also been demonstrated in single-cell recordings of Purkinje neurons (*Allan et al., 2015*; *Mason et al., 2006*), which may arise from cerebellar internal models of the dynamics of different types of objects (*Wolpert and Flanagan, 2001*; *Imamizu et al., 2000*; *Bursztyn et al., 2006*). Likewise, there is considerable evidence from human imaging studies and non-human primate neurophysiological studies for the role of prefrontal cortex and the striatum in perceptual category learning (*Freedman et al., 2001*; *Ashby and Maddox, 2005*; *Reber et al., 1998*; *Vogels et al., 2002*; *Seger and Miller, 2010*; *Antzoulatos and Miller, 2011*; *Antzoulatos and Miller, 2014*; *Bowman and Zeithamova, 2018*; *Raz and Saxe, 2020*), but it remains unknown whether these areas are also recruited in organizing objects based on their learned motor properties.

Current theories of motor learning often focus on graded generalization of learning across various stimulus and motor parameters as a revealing feature of the underlying computations (*Krakauer et al., 2000*; *Thoroughman and Shadmehr, 2000*; *Donchin et al., 2003*; *Ingram et al., 2017*). In particular, graded patterns of generalization have been taken as evidence that motor learning fundamentally involves associating contextual features of a movement with the target motor parameters in a continuous multi-dimensional space, often termed an associative map. The theoretical significance of our study is that it provides multiple, converging pieces of evidence for a fundamentally different type of organization—motor memories of objects are organized categorically, into families. Our key result is the family effect itself, wherein an outlier object is persistently encoded as a family member, despite greatly deviating from its expected weight. In contrast, the prediction of an associative map account is that these outliers would eventually be learned, since they are visually and haptically discriminable from the family (as shown by the accurate learning in the Uncorr+ condition).

In our experiments, we generally observed incomplete learning of the outlier when averaging anticipatory forces across participants. At first glance, partial learning could be explained by an associative map model where the neighboring objects reduce the estimated weight of the outlier by local generalization. However, seemingly partial learning is also consistent with the object families hypothesis. In particular, partial learning in the group averages could result from averaging together a subgroup of highly accurate learners with a separate subgroup of complete non-learners, who differ in their threshold for reclassifying the outlier as an individual. This latter interpretation was confirmed by our large-sample, web-based experiment, which revealed that individual differences in outlier learning followed an all-or-nothing pattern. At the end of the experiment, participants had either learned to classify the outlier as a unique individual and accurately estimated its weight, or they still encoded it as a family member and incorrectly estimated its weight based on the family representation.

We found remarkably similar results in the laboratory and web-based tasks. These two motor tasks are similar in that they both involve translating sensory information into continuous motor commands to achieve an action goal. However, the precise nature of the sensory information used for control differs between the tasks, with the laboratory task primarily relying on haptic feedback and the web-based task relying on visual feedback. Our web-based task required participants to map their weight predictions onto an arm movement that set the visual length of a spring. This 'visual' lifting is similar to many motor tasks in which the initial conditions of an interaction are likewise adjusted based on visual feedback; for example, when aiming in archery, lining up a putt in golf, or pulling back the plunger to launch a pinball. In these examples, the motor error is also provided strictly through visual feedback

as in our web-based task. The fact that we observe similar categorical encoding in both the laboratory task and the web-based task demonstrates the generalizability of our findings.

Our generalization results also favor a categorical organization of motor memory over a continuous, associative map. We found that the way that the outlier object was classified—either as a family member or an individual—had a dramatic effect on outlier-to-family generalization. When the outlier object was classified as a family member, strong generalization was observed, whereas when it was classified as an individual, generalization was substantially reduced. This qualitative change in generalization was observed across participants in different conditions, as well as within the same participants who, during learning, reclassified the outlier from a family member to an individual. These results strongly support the idea that motor memories of objects are organized categorically, rather than continuously, which would predict graded generalization as a function of error magnitude and sensory similarity. By eliciting separate visual classification of the outlier and the family objects, we were able to suddenly 'shut off' inter-object error generalization.

We also found that when the weight of the outlier was gradually increased from 1.2 to 1.5 kg, participants generally failed to learn its weight, even though it reached the same weight as the outlier that, when introduced abruptly, was learned. One interpretation of this finding is that first experiencing the 1.2 kg outlier, and then experiencing incrementally increasing weights, broadened the category by increasing the within-category variability, as shown in perceptual and conceptual categorization (*Rips, 1989*; *Huttenlocher et al., 2000*; *Clayards et al., 2008*). Another possible account for this finding is that category labels are 'sticky', and that once the test object was labeled as a family member, there was resistance to relabeling it as an individual, similar to the hysteretic effects reported in perceptual categorization (*Williams et al., 1986*; *Hock et al., 1993*; *Poltoratski and Tong, 2014*). However, it seems plausible that the 1.5 kg outlier was initially labeled as a family member as participants' anticipatory forces on the first lift of this object were based on the density of the family. If so, then relabeling occurred when this extreme outlier was learned, arguing against the 'stickiness' account. On the other hand, the stickiness hypothesis could account for the results we observed when the outlier weight was initially set to 1.5 kg and then gradually decreased to 1.2 kg. In this case, participants initially learned the extreme outlier and continued to accurately predict its weight—and hence to categorize it as an individual—even as its weight decreased to a level that, when introduced abruptly, was not learned. Alternatively, it is possible that learning the extreme 1.5 kg outlier as a distinct individual object caused the category boundary for the training objects to contract, such that a 1.2 kg outlier remained outside the learned family, perhaps because the individuated outlier effectively forms a competing category. Note that work on sensorimotor adaptation has shown that participants do not become aware of visual or force perturbations that are introduced gradually (*Kagerer et al., 1997*; *Malfait and Ostry, 2004*; *Klassen et al., 2005*; *Saijo and Gomi, 2010*; *Criscimagna-Hemminger et al., 2010*; *Roemmich and Bastian, 2015*). Since participants adapt to these gradually increasing perturbations, they never see large errors, which presumably explains why they do not become aware of the perturbation. In contrast, in our experiment with a gradually increasing outlier weight, participants did not adapt (i.e. they continued to predict the outlier weight based on the family density). Thus, they experienced larger and larger errors, ultimately experiencing the same error that drove learning when the 1.5 kg outlier was introduced abruptly. The reason that participants learned the 1.5 kg outlier when introduced abruptly, but not when introduced gradually, may be that they are sensitive to the change in error, as opposed to error per se.

Although the formation of motor memories has historically been viewed as a largely implicit process, recent research on motor learning and adaptation has emphasized the role of explicit processes. For example, when reaching under a visuomotor rotation, participants often learn to use an explicit re-aiming strategy to reduce movement errors (*Mazzoni and Krakauer, 2006*; *Anguera et al., 2010*; *Fernandez-Ruiz et al., 2011*), and can quickly recall and implement this strategy when re-exposed to the rotation at a later time (*Taylor et al., 2014*; *Huberdeau et al., 2015*). In the context of object manipulation, it is clear that people often have explicit knowledge of the weights of objects they interact with. That is, if asked, people can provide an estimate of the weight of an object. When lifting familiar objects, these estimates can be quite accurate, although research on weight illusions shows that these estimates are biased by expected weight (*Flanagan and Beltzner, 2000*; *Flanagan et al., 2008*). However, whether and, if so, how explicit knowledge is used when generating lift forces is unclear. In the current study, we did not ask participants to provide verbal estimates of the weights

of the objects before lifting them. We suspect that these estimates would have been consistent with the actual forces produced when lifting. Requiring participants to provide such estimates, however, may also alter the category boundary used to classify objects as family members or individuals. The use of explicit knowledge is believed to require working memory and previous research has shown that greater working memory resources are required when lifting unusually weighted objects than when lifting normally weighted objects (*Baugh et al., 2016*; *Flanagan et al., 2008*). Based on these findings, we speculate that when participants lift objects that are encoded as a family, this categorical encoding is largely implicit, requiring little to no explicit processing. In contrast, we suspect that explicit processing does contribute to encoding an object that is similar in appearance to the family as an outlier. Likewise, explicit processing may be required when learning the weights of multiple individual objects, as in our Uncorr+ condition. In this condition, response times were significantly greater than the Linear conditions, consistent with the notion that explicit processes were engaged. In general, as people become more experienced at classifying objects into separate families, the contribution of explicit knowledge will likely diminish and lifting will become more implicit and automated. As noted above, object manipulation tasks engage multiple sensorimotor and cognitive processes, including categorization, and can depend on implicit and explicit memories. From our perspective, any form of memory engaged in a motor control task can be considered as a 'motor memory', whether that is an explicit declarative memory or an implicit procedural memory.

By showing that dexterous object manipulation relies on learned representations of categories (and individuals), our findings open the door for future work that connects theories of human category learning, developed in the context of perception and cognition, with theories of motor control. The vast literature on category learning has identified and debated a variety of key issues, including why certain categorizations are harder to learn than others (*Ashby and Maddox, 2005*; *Shepard et al., 1961*), whether category knowledge is encoded using prototype, exemplar, or decision-bound representations (*Posner and Keele, 1968*; *Medin and Schaffer, 1978*; *Ashby and Townsend, 1986*), and how the relative contributions of explicit 'rule-based' and implicit 'information-integration' processes are modulated by the relevant perceptual dimensions and category structure of a stimulus domain (*Ashby and Maddox, 2005*; *Ashby et al., 1998*; *Ashby and Maddox, 2011*). A detailed review of how the pertinent findings from this literature might inform our understanding of dexterous object manipulation (and vice versa) is well beyond the scope of this article, but it is nonetheless clear that there is a pressing need for greater attention to these connections. However, focusing more narrowly on accounting for the present findings, it is notable that many existing process-level (i.e. trial-by-trial) models of category learning posit a mechanism that allows for the creation of a new category in memory when an observation deviates sufficiently from previously learned categories (*Ashby and Maddox, 2005*; *Hartigan, 1975*; *Carpenter and Grossberg, 1987*; *Clapper and Bower, 1991*; *Anderson, 1991*; *Love et al., 2004*; *Vanpaemel et al., 2005*). These various treatments can all be viewed as instances of non-parametric Bayesian models that leverage the hierarchical Dirichlet process, a statistically principled approach to clustering data into a theoretically infinite number of components (*Teh et al., 2005*; *Griffiths et al., 2007*). A recent motor control model has been developed based on this approach (*Heald et al., 2020*). However, at present this model cannot account for our results because there is no mechanism through which the visual properties of the objects—which are encoded as discrete cues—can be linked together to form a family.

In general, learning a family of objects based on covarying size and weight, as in this study, is presumably just one example of a more general tendency to compactly encode the covariability of observable sensory features and latent mechanical properties. Previous work has shown that people can learn more complex 'structures' in motor control tasks (e.g. visuomotor rotations and skews), but has not distinguished between categorical and associative representations (*Braun et al., 2009*; *Braun et al., 2010*). Categorical encoding amounts to carving the sparse, high-dimensional space of sensorimotor information into circumscribed, lower-dimensional object categories, providing a number of benefits. First, it allows for more robust interpolation and extrapolation from past sensorimotor experience by shoehorning ambiguous new items into predictable categories. Second, it reduces the temporal costs associated with specifically identifying objects, which would involve deeper traversal into object memory. Third, when working with multiple objects from the same family, this strategy conserves working memory resources that would otherwise be expended on object individuation. Lastly, categorical organization also conserves long-term memory resources by maintaining only

abstract descriptions of relevant family structure, rather than a detailed map of all sensorimotor properties, helping to address the curse of dimensionality. In contrast, although learning about individual objects may increase accuracy in some circumstances, this would come at the cost of significantly increased demands on attention (for visual recognition), cognitive control (for switching between memories), and memory (for storage). Therefore, in combination with context-sensitive reflexes and other rapid corrective mechanisms, a categorical memory of object properties affords tradeoffs between accuracy and memory that can be balanced as needed to support our unmatched ability to skillfully manipulate many different kinds of objects.

# Materials and methods

We first describe the in-laboratory experiments before describing the web-based experiments.

## Laboratory experiments

### Participants

A total of 80 participants (42 males, 38 females) aged 18–45 years old (median 24) were recruited for the laboratory experiments. Participants were right-handed according to the Edinburgh handedness questionnaire, and reported that they had normal or corrected-to-normal vision and no prior diagnosis of a movement disorder. They were compensated at a rate of $17 per hour. All experiments were conducted in accordance with the 1964 Declaration of Helsinki, following protocol approved by the Columbia University Institutional Review Board. Written informed consent was obtained from all participants prior to their participation.

### Apparatus

Experiments were performed using a 3BOT three-dimensional robotic manipulandum and an Oculus Rift DK2 (Menlo Park, CA) virtual reality headset, as well as a 2-button USB response pad (The Black Box ToolKit Ltd., Sheffield, UK). The position of the 3BOT handle was measured using optical encoders sampled at 5 kHz, and torque motors allowed forces (also updated at 5 kHz) to be generated on the handle. Participants sat on a height-adjustable stool in front of a tabletop workspace and grasped the 3BOT handle with their right hand (*Figure 1a*). The virtual reality headset was rigidly fixed to an aluminum crossbeam and angled downwards by 30°. Stereoscopic visual stimuli were rendered on the headset using custom OpenGL routines and the Psychophysics Toolbox (*Kleiner et al., 2007*). Auditory cues were provided through Sennheiser HD201 (Old Lyme, CT) over-ear headphones.

### Task

In our object 'lifting' task, the participant generates an upward force on an object that is initially fixed to the surface beneath it, such that the object cannot move. The participant then presses a button, at which time the surface disappears, releasing the object so that it is then free to move. The goal for the participant is to match the upward force to the weight of the object so that the object does not move when it is released. Participants performed this lifting task with five cylinders of equal radius (4.61 cm), but of different heights (6, 7.5, 9, 10.5, and 12 cm), leading to five equally spaced volumes (400, 500, 600, 700, and 800 cm$^3$). Each cylinder was shaded, from smallest to largest, between orange and red according to the Munsell color system (Hue: 10 R, Value/Chroma: 3/10, 4/12, 5/14, 6/16, and 7/16). All objects were visible throughout the task, except during rest breaks. The objects were positioned evenly around the edge of a gray, semi-transparent carousel with a radius of 20 cm (*Figure 1b*). The weight of each object varied across the experimental conditions (see below).

Before each trial, the 3BOT moved the participant's hand passively to a start position 11 cm in front of and 19 cm below the cyclopean eye (in gravity-oriented space) and clamped it there by a simulated stiff spring (spring constant: 4000 N m$^{-1}$, damping coefficient: 2 N m s$^{-1}$, both acting in all directions). The participant saw a stereoscopically rendered view of the five objects and the circular carousel (*Figure 1b*). The carousel rotated smoothly (750 ms) to bring a target object to the front and a 500 ms tone then signaled the start of the trial. Note that at this point, the hand (i.e. the center of the 3BOT handle) was located at the center of the base of the target object. The participant then generated an upward lifting force on the object (i.e. against the simulated stiff spring) attempting to match its weight. When ready, the participant pressed a button with their left hand that caused a portion of the

carousel below the object to open, thus releasing the object so that it was free to move. The physical interaction between the hand and the object was then simulated haptically using the 3BOT. We simulated the object as a point-mass acted upon by gravity and attached by a stiff, damped spring (acting in all three dimensions) to the center of the handle. The spring constant was 4000 N m$^{-1}$ and the damping coefficient was 2 N m s$^{-1}$ with gravity set at $-9.81$ m s$^{-2}$. We updated the location of the object both haptically and visually and generated the appropriate forces on the hand. This method produces a stable, compelling haptic percept of a handheld inertial mass. If the anticipatory force was more or less than the weight of the object then the handle would move upward or downward, respectively, until corrective motor commands re-stabilized the arm posture. To encourage accurate performance, thin horizontal gray bars (2 mm radius, purely visual and not haptic) were visible just above and below the target object from the start of the trial (not depicted in *Figure 1b*). If the object remained between the horizontal bars for 500 ms, the bars disappeared, and the participant completed the trial by raising the object at least 3 cm above the start position and replacing it on the carousel, where a virtual haptic surface was now simulated to allow full unloading of lift forces prior to the next trial. However, if the object crossed one of the bars, it turned red and a white-noise audio burst was played. The object had to be brought back within the bars before they would disappear, and only then could the participant complete the trial by raising and replacing the object on the carousel. The distance of the bars from the top and bottom edges of the object (i.e. the amount of tolerated object movement) varied according to the participant's performance: the demarcated region became 1 mm larger following an trial where the object crossed a bar, up to a maximum tolerated deviation of ±13 mm (this was also the initial width), and became 1 mm smaller after five consecutive trials where the object stayed within the bars, down to a minimum tolerated deviation of ±2 mm.

Feedback was also provided in the form of a per-trial score that depended on the absolute error between the anticipatory force at the moment of the button press and the required force to support the object, with score = max(0, 100–13*|error|). The participant's cumulative score was displayed throughout the experiment. The five highest-scoring previous participants' scores from the same condition were displayed in a leaderboard beside their own score. This leaderboard was initially seeded based on the score of a pilot run, which was multiplied by 1, 0.9, 0.8, 0.75, and 0.7 to produce five scores. These seed scores were erased one by one as data were collected from the first five participants in each condition.

## Paradigm

### Linear+ condition

Fifteen participants (of an initial sample of 30) were randomly assigned to the Linear+ condition; the other fifteen were assigned to the Uncorr+ condition (see below). The training objects (the two smallest and two largest objects by volume) weighed 600, 750, 1050, and 1200 g, respectively, corresponding to a constant density of 1.5 g cm$^{-3}$ (*Figure 1d*). The test object (or 'outlier') was the mid-size cylinder and weighed 1200 g, corresponding to a density of 2.0 g cm$^{-3}$.

All participants were informed that the purpose of the experiment was to test their ability to learn and recall the weights of a novel set of objects. The Linear+ condition began with a 120-trial training phase in which the participant interacted only with the four training objects. The order of presentation was pseudo-randomized in cycles where each object was presented once before any object was repeated, and subject to the additional constraint that the first object presented in one cycle could not be the same as the last object presented on the previous cycle. Following training, the test object (also called the outlier object when introduced amongst a linear object family) was introduced for a 200-trial test phase. During the test phase, in each five-trial cycle, the test object was always presented first, followed by the four training objects in pseudo-random order, but subject to the additional constraint that for every four cycles, each of the four training objects would be presented immediately after the test object (i.e. on the second trial of the cycle) exactly once.

To reduce the effects of fatigue, participants were required to take occasional 30 second breaks. During these breaks, participants stopped holding the 3BOT handle, came out of the virtual reality headset, and were encouraged to stretch their right arm and hand. These breaks occurred after trials 60, 120, and 200. The experiment had a total of 320 trials and lasted approximately 45 minutes.

Prior to the experiment, the experimenter demonstrated the task by performing 10 or 15 trials of a familiarization condition while the participant watched. The visual scene was displayed on a nearby

monitor so the participant could follow along. The participant then completed 30 trials of task familiarization, where the object stimuli were three spheres (5 cm radius) that were blue, red, and green (7.5B 6/8, 7.5 R 6/18, 7.5 GY 6/10) and weighed 500, 900, and 1300 g, respectively. During task familiarization, the experimenter could choose to display or hide a bar graph that showed the real-time load force on the handle. This visual aid helped participants calibrate to the range of forces they would be asked to produce in the experiment, and prevented them from producing unnecessarily large forces. Approximately ten familiarization trials were performed with full view of this visual feedback, followed by approximately ten trials with short glimpses of the feedback prior to the button press, followed by approximately ten trials without the visual feedback as in the actual experiment.

### Uncorr+ condition

Fifteen participants were randomly assigned to the Uncorr+ condition. The Uncorr+ condition was similar to the Linear+ condition, except the four training object weights (600, 750, 1050, and 1200 g) were assigned randomly to the four training objects (*Figure 1f*), subject to the constraint that the absolute value of the Pearson correlation coefficient between volume and mass could not exceed 0.3. The test object had the same weight as in the Linear+ condition.

### Linear++ condition

In the Linear++ condition, we recruited participants until we obtained a sample size of 9 after excluding non-learners. The Linear++ condition was identical to the Linear+ condition, except the outlier object weighed 1500 g (rather than 1200 g; *Figure 1e*).

### +Linear condition

In the +Linear condition, we recruited participants until we obtained a sample size of 11 after excluding non-learners. In the +Linear condition, the experiment began with a 30-trial training phase where participants interacted only with the test object which weighed 1200 g. This was followed by a 200-trial test phase identical to the Linear+ condition in which all five objects were lifted in each cycle. This was followed by the 1:1 phase, which was a block of 10 cycles where, in each cycle, the test object was presented four times and each of the four family objects was lifted once, for a total of ight trials per cycle. To limit the number of consecutive presentations of the test object in the 1:1 phase, we pseudo-randomized the trial sequence such that consecutive presentations of the test object occurred exactly 13 times, while presentations of the test object with one, two, or three intervening trials from the last presentation of the test object occurred exactly 15, 8, and three times, respectively. The +Linear condition had a total of 310 trials and rest breaks occurred after trials 90 and 190.

### ++Linear condition

The ++Linear condition was identical to the +Linear condition except the outlier object weighed 1500 g (rather than 1200 g).

### Linear↗ condition

In the Linear↗ condition, we recruited participants until we obtained a sample size of 9 after excluding non-learners. The Linear↗ condition was identical to the Linear+ condition except that the outlier object's weight (initially 1200 g) was iteratively increased by 50 g on trials 221, 261, 301, 341, 381, and 421, up to a maximum of 1500 g. The length of the test phase was also increased to 340 trials, leading to a total of 460 trials. Rest breaks occurred after trials 60, 120, 220, 300, and 380.

### Linear↘ condition

The Linear↘ condition was identical to the Linear↗ condition except that the outlier initially weighed 1500 g and its weight was iteratively decreased by 50 g to 1200 g.

## Analysis

### Data preprocessing

The anticipatory force was taken as the average force applied in the upward direction over the final ten samples (10 ms) of the clamp phase (*Figure 1c*, trial phase 2). Response times were measured

as the duration from trial onset (defined as the beginning of trial phase 2, when the object carousel stopped rotating) to the button press.

We excluded 322 anticipatory forces (1.15%) that were less than or equal to 1 N (typically due to an accidental button press) or more than 3.5 scaled median absolute deviations away from the median anticipatory force applied by a given participant for a given object. Similarly, we excluded 392 response times (1.40%) that, following a log transformation, were more than 3.5 scaled median absolute deviations from the median log-transformed response time. We then imputed the mean anticipatory force or reaction time produced on non-outlying trials by other participants for the same object, cycle, and condition.

We also excluded participants (and hence recruited additional participants) who failed to learn the weights of the training objects, as the goal of the experiment was to observe how learning of a new object is affected by existing knowledge of object weights. Non-learners were defined as those whose anticipatory forces during the final 15 cycles of the training phase did not show a highly significant ($\alpha$ = 0.01) positive correlation with the weights of the objects. In the Uncorr+ group, three participants were excluded by this criterion. In the Linear+ and Linear↗ groups, one participant from each group was excluded by this criterion. This criterion was not applied in the +Linear and ++Linear groups because the training phase involved only the test object.

## Statistical analysis

In most motor learning experiments, there are between eight and twelve participants per experimental group. This sample size provides sufficient power to detect the large effects typical of motor learning experiments, where the effect of interest is observed in most if not all participants. As this was a new experimental paradigm, in the first two experimental groups (Linear+ and Uncorr+) we recruited a sample size of fifteen. In the Uncorr+ group, we observed significant learning of the outlier object with a large effect size (Cohen's $d$ = 1.17). Based on this value, we adopted a sample size of nine for the Linear++, Linear↗, and Linear↘ groups, aiming to achieve a statistical power exceeding 0.90 in our one-tailed $t$-tests of outlier learning. In the +Linear and ++Linear conditions, we could not exclude individual participants as non-learners as in the other conditions (see above). We therefore estimated a slightly reduced effect size for sample size estimation (Cohen's $d$ = 1.00), leading us to adopt a sample size of eleven in order to achieve at least 0.90 power in these groups. Post-hoc power analyses of groups with significant outlier learning confirmed that we achieved the desired power (Uncorr+: 0.98, Linear++: 0.92, ++Linear: 0.96, Linear↘: 0.99).

In the Linear+, Linear++, Uncorr+, Linear↗, and Linear↘ groups, learning of the training set at the end of the training phase was measured using the Pearson correlation between actual object weight and anticipatory force on trials between trial cycles 23 and 30. The Fisher $z$-transformation was used to compute 95% confidence intervals.

To assess learning of the test object relative to the training objects, we compared the anticipatory force for the test object to the force that would be expected based on the anticipatory forces for the four training objects (i.e. the 'family-predicted' weight). To do this, we fit a linear regression to the anticipatory forces for the training objects as a function of volume in the final 16 trial cycles of the test phase. We calculated the family-predicted weight of the test object based on the regression and the test object's volume. Note that because the test object's volume was always in the middle of the training objects, the family-predicted weight is equivalent to the mean anticipatory force produced for the four training objects, hence the logic is also appropriate for the Uncorr+ condition. We used one-tailed $t$-tests to evaluate the null hypothesis that the test object weight would not be learned. One-tailed tests are justified because failure to learn the test object weight is a directional hypothesis, which includes the case where the anticipatory force for the test object does not differ from the family-predicted weight, as well as the case where it is less than the family-predicted weight. In the Linear↗ and Linear↘ groups, we also conducted this analysis for the final four trial cycles of the initial portion of the test phase during which the test object weight did not change.

In the first experiment, we analyzed how lifting the test object generalized to the training objects in the subsequent trial (*Figure 2g–i*). For this analysis, we focused only on trials that immediately followed a lift of the test object (i.e. the second trial of each trial cycle), as any generalization would only be reduced in the third, fourth, and fifth trials of each cycle due to washout from lifting the training objects. Generalization was measured by comparing the anticipatory force on these trials,

during either the early test phase (trial cycles 31–34) or the end of the test phase (trial cycles 55–70), with the average anticipatory force at the end of the training phase (trial cycles 27–30), and expressed as a percentage of the difference between the actual and expected weight of the test object (2.94 N in Linear+ and Uncorr+ versus 5.89 N in Linear++). Due to the constrained trial order in the test phase (see Paradigm), all four training objects contribute equally to this analysis. For each group, we conducted two-tailed $t$-tests on this generalization metric in each portion of the test phase, and also on the change in this generalization metric from the early portion to the late portion of the test phase.

In the first experiment, we conducted a two-way repeated-measures ANOVA on log-transformed response times, with factors Group (two levels: Linear+ combined with Linear++ versus Uncorr+) and Epoch (four levels: trial cycles 1–15, 16–30, 31–50, 51–70), and performed four follow-up one-tailed $t$-tests to examine whether the main effect of Group was present in all four Epochs individually. In each of these groups, we also tested for single-trial generalization at the start (first four cycles) and the end (final sixteen cycles) of the test phase, as well as the change from start to end, using two-tailed $t$-tests. In the Linear++ group, where we observed learning of separate categories, we analyzed whether response times were longer on trials involving a category switch (i.e. trials with the test object or trials immediately after the test object). Not surprisingly, participants in our experiments took longer to generate larger anticipatory forces. To account for this uninteresting component of the response time, we fit a linear regression to response time as a function of object weight using data from 'non-switch' trials in the Linear+ and Linear++ conditions (i.e. excluding trials with the test object and immediately subsequent trials). This allowed us to determine the predicted response time, based solely on weight, which we then compared to the actual response time to test for additional temporal costs (e.g. associated with switching categories). We found a significant slope of 26.2 ms per Newton of anticipatory force. Thus, when analyzing test object trials in the Linear++ condition, we computed the predicted response time as the average response time for the family objects in 'non-switch' trials during the test phase plus 154 ms, as the test object was 5.89 N heavier than the average family object weight. When analyzing trials immediately after the test object, we computed the predicted response time as the average response time for the family objects in 'non-switch' trials during the test phase, as the object weight in these trials was, on average, equivalent to the average family object weight.

We also directly compared the Linear↗ with the Linear++ group, and the Linear↘ with the Linear+ group, using two-tailed, two-sample $t$-tests on the anticipatory force for the test object in the final 16 trial cycles of the test phase, when the outlier weight was similar for each pair of groups.

## Web-based experiment

For the web-based experiments, we obtained complete data associated with 196 unique Amazon Mechanical Turk Worker IDs (135 males, 60 females, 1 non-binary) aged 19–70 years old (median 31.5). These workers were paid $1.50 upon successful submission of a complete dataset, and received an additional bonus payment determined by dividing their final score by 100 (max bonus = $0.01/trial = $1.60). Of these participants, 185 individuals reported using their right hand to control their input device and 11 reported using their left hand. They were not screened for visual impairment or prior diagnosis of movement disorder.

The web-based experiments were designed so that they could only be completed by individuals using the Google Chrome web browser, in full-screen mode and with pointer lock enabled, on a computer with graphics hardware that supports WebGL 2.0, and with a mouse (172 participants) or trackpad (24 participants). Dimensions of the full-screen window displaying the task ranged from (1093, 576) to (2560, 1410) pixels; actual monitor sizes were not collected.

The objects in the web-based experiments had radii of 2 cm and heights of 3, 4, 5, 6, and 7 cm. They were arranged around a gray metallic ring, had springs attached to their tops, and were rendered via perspective projection to a camera 40 cm behind and 10 cm above the top-center of the foremost object. Since there was no haptic interface, feedback about object weight was provided through vision of the simulated dynamics of a spring-mass-damper system (*Figure 5a*). In the web-based Linear++, Linear+, Linear-, and Linear-- conditions, the training objects always weighed 300, 400, 600, and 700 g, while the test object weighed 900, 700, 300, or 100 g, respectively.

Trials of the web-based experiments were similar to the laboratory experiment, but simplified. There were no auditory cues, haptic feedback, bars above and below the object, or a leaderboard. Each trial consisted of two main phases (*Figure 5a*): the clamp phase (trial phase 1), in which the

participant clicked and dragged to stretch the spring on top of the object, and the release phase, which was triggered by pressing the Shift key with the spring stretched to a certain distance, and portrayed a simulation of the spring-mass-damper dynamics that would result from the initial conditions created by the spring length (spring constant: 1, damping coefficient: 0.01). The per-trial score $y$ was related to the spring-length error in centimeters $e$ by $y = \max(0, 1-e^2/2.25)*100$. The duration of the release phase in seconds $t$ (i.e. the inter-trial interval, which serves as a time penalty) was modulated according to the spring-length error: $t = \min(0.4*e^2, 12)$. This time penalty was correlated with, but not exactly equal to, the decay time of the oscillations in the visual feedback of the spring.

Participants received task familiarization through a single, repeatable demo trial that provided an instructed walkthrough of a single trial with the largest of the four training objects. The total number of trials was reduced by half compared to the in-laboratory Linear+ condition, with 60 training trials and 100 test trials. Rest breaks were not required.

The anticipatory force was measured as the amount of force exerted on the object by the visually simulated spring on the final frame of the clamp phase (*Figure 5a*, trial phase 1). Non-learners were defined as those whose anticipatory forces for the training objects during the final five cycles of the training phase *or* the final five cycles of the test phase did not show a mild positive correlation with the simulated weights ($\alpha = 0.10$). Forty-seven participants were excluded from the four groups of the web-based experiment by this criterion, resulting in sample sizes of 37, 36, 37, and 39 individuals, respectively, in the Linear++, Linear+, Linear-, and Linear-- groups. This high rate of exclusion was not due to task difficulty, but to the fact that many participants in the web-based experiment adopted strategies that minimized effort at the expense of time and accuracy. Additionally, we excluded as outliers any anticipatory forces that were more than four scaled median absolute deviations from the median anticipatory force applied by a given participant to a given object, resulting in 1398 exclusions (4.46%).

To estimate required sample sizes for the web-based experiments, we simulated bimodal distributions of 'learners' and 'non-learners' with different sample sizes and calculated the proportion of simulations in which the two-Gaussian mixture model outperformed the single Gaussian model. We estimated that the learner and non-learner group means would be separated by 3.5 standard deviations, and we assumed that learners and non-learners are normally distributed, have equal variance, and occur in equal proportions. We found that a sample size of 36 participants led the two-Gaussian model to be correctly favored by AIC in 85 % of our simulations.

We analyzed the distributions of anticipatory forces produced for the outlier in the final five cycles of the test phase. We fit both a single-Gaussian and a two-Gaussian mixture model using the R package *mclust* (*R Development Core Team, 2020*; *Fraley and Raftery, 2002*), and estimated confidence intervals on the fit parameters by parametric bootstrap with 10,000 samples. Model comparisons based on AIC and BIC yielded the same pattern of results; we report only AIC in the text. The same analysis of bimodality was conducted for the laboratory experiments by combining the Linear+ and +Linear groups in one analysis, and the Linear++ and ++Linear groups in another analysis. For the laboratory experiments, we fit the single-Gaussian and two-Gaussian models to the distributions of the difference between anticipatory force for the outlier and the family-predicted weight of the outlier in the final 16 trial cycles of the test phase.

All source data, analysis code, and figure generation code is available in the supplementary files.

## Acknowledgements

We thank Ian Howard for the design of the 3BOT manipulandum.

## Additional information

### Funding

| Funder | Grant reference number | Author |
| --- | --- | --- |
| National Institutes of Health | R01NS117699 | Daniel M Wolpert |

| Funder | Grant reference number | Author |
| --- | --- | --- |
| National Institutes of Health | U19NS104649 | Daniel M Wolpert |
| Natural Sciences and Engineering Research Council of Canada | | J Randall Flanagan |
| Canadian Institutes of Health Research | | J Randall Flanagan |

The funders had no role in study design, data collection and interpretation, or the decision to submit the work for publication.

## Author contributions

Evan Cesanek, Conceptualization, Formal analysis, Investigation, Methodology, Project administration, Software, Validation, Visualization, Writing – original draft, Writing – review and editing; Zhaoran Zhang, Writing – review and editing; James N Ingram, Resources, Software, Writing – review and editing; Daniel M Wolpert, Conceptualization, Methodology, Resources, Supervision, Writing – original draft, Writing – review and editing; J Randall Flanagan, Conceptualization, Methodology, Supervision, Writing – original draft, Writing – review and editing

## Author ORCIDs

Evan Cesanek http://orcid.org/0000-0002-5335-6604
Zhaoran Zhang http://orcid.org/0000-0002-4192-4088
James N Ingram http://orcid.org/0000-0003-2567-504X
Daniel M Wolpert http://orcid.org/0000-0003-2011-2790
J Randall Flanagan http://orcid.org/0000-0003-2760-6005

## Ethics

Human subjects: All experiments were conducted in accordance with the 1964 Declaration of Helsinki, following protocol approved by the Columbia University Institutional Review Board (IRB-AAAR9148). Written informed consent was obtained from all participants prior to their participation.

## Decision letter and Author response

Decision letter https://doi.org/10.7554/eLife.71627.sa1
Author response https://doi.org/10.7554/eLife.71627.sa2

# Additional files

## Supplementary files

• Transparent reporting form

## Data availability

All source data, analysis code, and figure generation code is available in the supplementary files.

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
