## [Editor Report]

This paper provides compelling evidence from several behavioural experiments that recently learned estimates of the mass of novel objects possess a categorical structure in memory. It further links this categorical structure to important aspects of motor control, and provides a compelling window through which to consider the role of multiple systems in the learning and memory of novel object dynamics.

---

## [Decision Letter]

**Decision letter after peer review:**

Thank you for submitting your article "Motor memories of object dynamics are categorically organized" for consideration by *eLife*. Your article has been reviewed by 3 peer reviewers, including Matthew Crossley as Reviewer #1 and Reviewing Editor, and the evaluation has been overseen by Richard Ivry as the Senior Editor.

Essential revisions:

We all agree that the experiments are well designed and well executed, and the results confidently point to the presence of a categorical effect. However, we also agree that the points discussed below represent essential revisions.

1. We are unsure how clearly these data can be anchored to motor learning and memory. R2 points out that the categorical effect could arise from declarative memory and R3 suggests that the categorical effect could arise from a combination of map-learning + heuristic category judgement (which would likely be declarative or otherwise non-motor).

This raises a fundamental question: Does this paper demonstrate a categorical structure to motor memory, or is it telling us something about non-motor cognition?

Please clearly address this possibility in your revision. In so doing, please explicitly state your views on how motor or non-motor both the lab-based and the web-based task are, and please also state to what degree you feel the web-based task is measuring the same phenomena as the lab-based task.

One observation that may be helpful in considering how to structure your revision is that since the family bias appears to effectively block learning of the true weight, it seems that even if the categorical effect is driven by non-motor systems, it can nevertheless have an important impact on motor systems.

2. If we accept that the categorical effect is coming from motor systems, then the degree to which it is novel and / or counter to the predictions of leading models should be made clearer. For example, my own intuition is that the categorical effect could arise from switching between "multiple paired forward and inverse models for motor control" (Wolpert and Kawato, 1998). I believe that this would be best addressed by adding formal modelling to your paper, but I am open to the idea that simply rewriting may be sufficient.

3. The basic model predictions are reasonably clear as is, but I agree with R3 that it isn't as clear as it could be whether or not block-level generalisation effects should exist in your data. Again, adding formal modelling would strongly address this ambiguity, but may not be necessary. If block level generalisation effects are predicted, then please report them in your revision.

4. R2 and R3 both raise the issue that there are useful reaction time predictions that should be investigated. In essence, we should see RT switch costs wherever a categorical effect is established and not elsewhere. In the revision, please test these predictions. If they are invalid, or the analysis isn't a good idea for some other reason, please explain why.

5. R2 and R3 both raise issues related to whether the lab-based task was bimodal. Please report this analysis in your revision, or else clearly outline why it isn't appropriate to do so (e.g., insufficient sample size etc.).

*Reviewer #1 (Recommendations for the authors):*

I have only two major recommendations for the authors.

1. The paper is currently framed by asking "how memories of multiple objects might be organized", but I believe the paper would be clearer by using the following two passages from the discussion as anchors:

"Current theories of motor learning often focus on graded generalization of learning across various stimulus and motor parameters as a revealing feature of the underlying computations 38,97-99. In particular, graded patterns of generalization have been taken as evidence that motor learning fundamentally involves associating contextual features of a movement with the target motor parameters in a continuous multi-dimensional space, often termed an associative map. The theoretical significance of our study is that it provides multiple, converging pieces of evidence for a fundamentally different type of organization-motor memories of objects are organized categorically, into families"

"These various treatments can all be viewed as instances of non-parametric Bayesian models that leverage the hierarchical Dirichlet process, a statistically principled approach to clustering data into a theoretically infinite number of components 123,124. Importantly, this approach has recently been applied to successfully account for an unprecedented range of phenomena in motor learning 125, suggesting that similar computations could also underlie the (in)ability to learn the weight of an outlier object in our lifting task."

I think this framing would give you the opportunity explicitly walk the reader through the modelling of Wolpert and Kawato (1998) -- and any more recent motor learning theories that carry on in the internal model tradition and / or the context inference tradition -- such that we understand right from the beginning how the existence of categorical encoding would demand revision of our current thinking. If it turns out that the present data are completely consistent with current models, then you could use this framing to explain why the present data is nonetheless important. This would presumably entail a cogent explanation as to why the present data is itself novel. That is, I would like a clear description of the context effects studied in force fields and in any other relevant motor task such that the categorical effect you report can be seen to stand in sharp contrast.

2. I believe the strength of the paper would be greatly improved if you performed modelling showing how and why the ideas of Wolpert and Kawato (1998) and related models are inadequate to capture the present data.

Wolpert, D. M., and Kawato, M. (1998). Multiple paired forward and inverse models for motor control. Neural Networks, 11(7-8), 1317-1329.

*Reviewer #2 (Recommendations for the authors):*

I would like to raise several issues.

1. In figure 2a (Linear+ group), the data of the 1st test trial (31st cycle) for the test object located in the just middle of the four training objects (i.e., ~9N). This result was reasonable from the idea of family effect: The motor system might estimate the weight by the interpolation. However, in figure 2c (Linear++ group), the participants failed to estimate the weight for the test object from the 1st test trial. This result was puzzling. Is there any reasonable explanation? Was there any possibility that this failure triggered to develop the different motor memory?

2. Line 280-282 "These results suggest that….": If the processing objects in a family contribute to reducing the response time, I expect that the response time for the test object in Linear++ group should also exhibit a longer response time. Did the results support this prediction?

3. Line 390- "All-or-nothing learning of outlier weight": The results of the web-based task clearly indicated that the estimated weight distribution for Linear++ or Linear-- group was bimodal. Was bimodality also observed in the laboratory task? The number of participants for the Linear++ and ++Linear groups was sufficient to examine if the distribution was bimodal.

4. The web-based task: This is an innovative way of collecting data from many subjects. However, I wondered if this web-based task can be regarded as a motor learning task. In this task, the participants could directly see the motor output by the spring length. This was totally different from the laboratory experiment in which the participants could only use their haptic sensation. Considering that the participants had only to associate the object size with the spring length, the web-based task was like a test for declarative memory. If the web-based task was not a motor task, the object family effect could not be a property of motor memory but a property of declarative memory.

5. I was curious about how the participants recognized the error when lifting the test object. The difference in error awareness between Linear++ and Linear+ groups contributed to either creating a new memory or consistently using the same memory. I would also like to know if the participants noticed gradual weight change in Linear_up and Linear_down groups (Figure 4).

*Reviewer #3 (Recommendations for the authors):*

In this paper, Cesanek et al., use a novel object lifting task to investigate the "format" of memories for object dynamics. Namely, they ask if those memories are organized according to a smooth, local map, or discrete categories ('families'). They pit these competing models against one another across several experiments, asking if subjects' predicted weights of objects follow the family model or a smooth map. This was tested by having people train on objects of varying volumes/masses that were either consistent with a linear mapping between volume/mass, or where those dimensions were uncorrelated. This training phase was either preceded by, or preceded, a testing phase where a novel object with a deviant mass (but a medium-size volume) was introduced. As the authors expected, individuals trained on the linear mappings treated novel objects that were relatively close to the "family" average mass as a member of the family, and thus obligatorily interpolated to compute the expected mass of that object (i.e., under-predicting its true mass); conversely, when a novel object's mass was a substantial outlier w/r/t the training items, it was treated as a singleton and thus lifted with close to the correct force. Additional variations of this experiment provided further evidence that people tend to treat an object's dynamical features as a category label, rather than simply forming local associative representations. These findings offer a novel perspective on how people learn and remember the dynamics of objects in the world.

Overall, I found this study to be both rigorous and creative. The experimental logic is refreshingly clear, and the results, which are replicated and extended several times in follow-up experiments, are rather convincing. I do think some additional analyses could be done, and data presentation could be improved. I also thought the generalization analysis, as I interpreted it, was difficult to align with the initial predictions.

I think sequential effects in the test phase could be analyzed more closely. This is especially pertinent as the generalization metric appears to be based primarily on sequential effects (if I'm interpreting it correctly). Moreover, wouldn't the family model predict a block-level generalization effect? That is, if the family representation is a discrete memory, and is altered by incorporating the new, heavier object, wouldn't all items be influenced at a block-level, not just items nearby in volume/time? I think overall the connection between the family model and the particular implementation of the generalization analysis didn't fully click for me.

I had trouble figuring out why there are different predictions for the family/association models in Figure 1F?

Shouldn't response times for the Linear+/Linear++ conditions be different when subjects are faced with the novel object? (i.e., a memory "switch" cost?). This is hard to see here as they have been collapsed. Moreover, it could be useful to also analyze response times from stimulus appearance to initial force application (planning?).

The consistent undershoot in the novel object's mass for the Linear++/++Linear groups is treated, retrospectively, as the result of individual differences rather than a "true" undershoot (i.e., via the online experiment). Could this be confirmed in the original in-lab samples? I understand they are smaller Ns; at least it could be nice to see the individual-level data if the model fit is underpowered. (Indeed, individuals could be added to several of the summary statistic figures).

It seems plausible that the computation of an anticipatory force could come from parallel learning processes (e.g., very slow local map-learning + heuristic category judgment) within a single observer. Couldn't this also lead to the observed undershoot in the Linear ++ condition? While not critical, I think this point could be discussed somewhere.

---

## [Author Response]

Essential revisions:We all agree that the experiments are well designed and well executed, and the results confidently point to the presence of a categorical effect. However, we also agree that the points discussed below represent essential revisions.1. We are unsure how clearly these data can be anchored to motor learning and memory. R2 points out that the categorical effect could arise from declarative memory and R3 suggests that the categorical effect could arise from a combination of map-learning + heuristic category judgement (which would likely be declarative or otherwise non-motor).This raises a fundamental question: Does this paper demonstrate a categorical structure to motor memory, or is it telling us something about non-motor cognition?Please clearly address this possibility in your revision. In so doing, please explicitly state your views on how motor or non-motor both the lab-based and the web-based task are, and please also state to what degree you feel the web-based task is measuring the same phenomena as the lab-based task.One observation that may be helpful in considering how to structure your revision is that since the family bias appears to effectively block learning of the true weight, it seems that even if the categorical effect is driven by non-motor systems, it can nevertheless have an important impact on motor systems.

The reviewers ask us to discuss whether the categorical memory organization we observe for object weights should be viewed as motor memory or non-motor (or ‘cognitive’) memory. We feel that, ultimately, memory can be designated ‘motor memory’ if it is used to control a motor task. This includes explicit or declarative memories that are engaged in motor tasks. Consider, for example, the explicit re-aiming that is often observed when adapting to a visuomotor rotation. It seems clear that re-aiming draws on similar explicit processing as mental rotation (*i.e.*, mentally rotating the visual target around the starting position of the hand). However, it is not obvious to us that this processing should necessarily be characterized as non-motor. Rather, it is a capacity that can support both motor and non-motor tasks. Similarly, while categorical encoding of object weights (and mechanical properties more generally) is clearly important for motor control, it is conceivable that it may also be important in non-motor tasks (*e.g.*, tasks involving perception or visual cognition without action). Again, even if such encoding was used in a non-motor task, it is not clear to us why we would label such encoding—in general—as being strictly non-motor. In the revised Discussion, we now clarify our view on why the categorical encoding we observe is related to the organization of motor memory:

L677: “As noted above, object manipulation tasks engage multiple sensorimotor and cognitive processes, including categorization, and can depend on implicit and explicit memories. From our perspective, any form of memory engaged in a motor control task can be considered as a ‘motor memory’, whether that is an explicit declarative memory or an implicit procedural memory.”

The reviewers also ask whether the categorical encoding we observe involves explicit (or declarative) memory or implicit (or procedural) memory. We know from research in other domains that categorical encoding may be explicit or implicit. For example, researchers who work on categorization distinguish between rule-based categories, thought to be learned explicitly, and information-integration categories, thought to be learned implicitly.

In our experiment, we did not attempt to measure whether the generation of lift forces involves explicit processes, implicit processes, or both. Although we can speculate about the processes that might be involved, we feel that this is a challenging issue. We know that if we ask people to verbally estimate the weights of objects before lifting them, they can generally give reasonable answers. However, we also know that these explicit judgments are not estimates of *absolute* object weight, but are biased by expected weight (as clearly demonstrated by the size-weight illusion). In other words, explicit weight estimates are partially comparative. We suspect that if we had asked participants who did not learn the outlier to explicitly estimate its weight prior to lifting, they would have given responses consistent with the family-predicted weight. Conversely, we suspect that participants who did learn the outlier would have given estimates close to the actual weight. That is, we think that participants would likely exhibit explicit knowledge of the outlier in cases where correct lift forces are learned for the outlier. (Note that we also suspect that asking participants to provide verbal estimates of weight might shift the family boundary; *e.g.*, participants might be more likely to generate new categories if asked to reflect on the weight.)

Even though people can give explicit (albeit biased) estimates of object weight, it is unclear how or whether this explicit knowledge is used to generate forces when lifting. The use of explicit knowledge to perform a task is often considered to require working memory resources, whereas the use of implicit knowledge is more automated and does not require working memory resources. Thus, one way to probe the involvement of explicit memory in object lifting would be to consider the working memory resources required to perform the task. We have previously provided evidence that lifting unusually weighted objects taxes working memory resources (Baugh et al., 2016, J. Neurophysiology). Specifically, having to perform a secondary task that draws on working memory disrupts predictive scaling of lift forces when lifting unusually weighted objects, but not when lifting normally weighted objects.

Based on these findings, we speculate that when participants lift objects that are all encoded as members of the same family, this categorical encoding occurs with very little engagement of explicit processing. In contrast, we suspect that explicit processing contributes to encoding an object that is similar in appearance to the family as an outlier. In particular, explicit processing may be required to inhibit automatic category encoding and recognize that the object does not belong to the family. It is possible that this explicit processing could be reduced under some conditions, such as with extended practice, by introducing additional objects that form a family with the outlier, and/or with greater visual discriminability between the family objects and the outlier.

In order to clarify these issues in our manuscript, we have considerably modified the paragraph in the Discussion that deals with explicit and implicit contributions to object lifting (red text is new):

L654: “Although the formation of motor memories has historically been viewed as a largely implicit process, recent research on motor learning and adaptation has emphasized the role of explicit processes. […] In general, as people become more experienced at classifying objects into separate families, the contribution of explicit knowledge will likely diminish and lifting will become more implicit and automated.”

Finally, the reviewers ask whether the web-based task tests the same phenomena as the lab-based task, and Reviewer 2 questions whether the former is really a motor learning task. For us, there is no question that both tasks are motor learning tasks. Both tasks examine how individuals translate sensory information into continuous motor commands to achieve an action goal. Although the precise nature of the sensory information used for control differs between the tasks, both tasks engage similar control mechanisms.

In our web-based task, participants’ weight predictions were mapped onto the visual length of a spring (rather than a haptic sensation). However, there are many motor tasks in which the initial conditions of an interaction are adjusted based on visual feedback, like when aiming in archery, or lining up a putt in golf. In these examples, the motor error is also provided strictly through visual feedback as in our web-based task.

In the revised paper, we have added a sentence when introducing the web-based tasks that emphasizes this similarity:

L442: “Note that although the exact nature of the sensory information in the web-based task differs from the laboratory task, both are fundamentally motor control tasks because they test how individuals translate sensory information into continuous motor commands to achieve an action goal.”

We have also added the following paragraph to the Discussion:

L606: “We found remarkably similar results in the laboratory and web-based tasks. […] The fact that we observe similar categorical encoding in both the laboratory task and the web-based task demonstrates the generalizability of our findings.”

2. If we accept that the categorical effect is coming from motor systems, then the degree to which it is novel and / or counter to the predictions of leading models should be made clearer. For example, my own intuition is that the categorical effect could arise from switching between "multiple paired forward and inverse models for motor control" (Wolpert and Kawato, 1998). I believe that this would be best addressed by adding formal modelling to your paper, but I am open to the idea that simply rewriting may be sufficient.

We have carefully considered how our results fit with modular models of motor control including Wolpert and Kawato’s (1998) MOSAIC model and we have concluded that such models cannot explain our data. In what follows, we explain why the MOSAIC model cannot account for our results and then provide a simulation, based on the model, demonstrating this finding. The MOSAIC model assumes that the motor system has at its disposal an effectively unlimited supply of learning modules (*i.e.*, paired forward and inverse models) that will be used as needed. To be concrete, with respect to the Linear+ experiment, the MOSAIC model should recruit four learning modules, each primarily responsible for one of the four unique weights (recall that in Linear+ the outlier object weighs the same as the largest training object, so there are five objects but four weights). This is because the modules become differentiated only by having different forward model predictions and, in our task, these predictions depend only on weight. As a result, each of the four modules would learn to predict a high responsibility for one of the four weights, and each module’s inverse model would simply generate the motor command equal to the weight that is well predicted by its paired forward model. Note that when the outlier is introduced later in the experiment, it will already be well predicted by one of these modules, as it shares the same weight as another object.

However, producing the correct motor command for the test object first requires that the responsibility predictor for this module learns to predict a high responsibility based on the observed sensory cue. In our task, the only relevant sensory cue was the height of the object. At the start of each trial, the responsibility predictors determine the weighting of the inverse model outputs of all the modules shaping the final motor command. Over time, each responsibility predictor learns to predict, from sensory cues alone, the responsibility based on the forward model errors, so high responsibility is predicted for modules with low forward model error. Initially, the responsibility predictor for the module that is already tuned to the weight of the outlier will not predict a high responsibility because it was trained on the largest height, whereas the outlier is medium height. Note that in order to learn to predict the same high responsibility for the medium and largest heights, this responsibility predictor must implement a non-linear mapping, which should be possible as neural networks are universal function approximators. The Uncorr+ condition suggests that in the MOSAIC framework, at least four modules can be recruited, learned and accessed by height (*i.e.*, the responsibility predictor can implement a highly nonlinear mapping).

Thus, a MOSAIC model, if trained properly to reach the optimal solution with minimum average error across the task, would learn the weights of all objects, failing to account for our main finding. Indeed, we show in Author response image 1 that this is exactly what happens when a MOSAIC model is given our task.

**Author response image 1. sa2fig1:** A MOSAIC model learns the weight of the Linear+ outlier in 200 trials. Our model consists of five learning modules, each containing a responsibility predictor (a 1-10-1 feedforward neural network), an inverse model (with one scalar parameter as the motor output, *i.e.* the anticipatory force), and a forward model (with one scalar parameter as the predicted weight, subtracted from the final weighted motor output to predict the observed outcome, *i.e.* the force error). We first trained the model on the four training objects, which caused four of the five learning modules to learn each of the four different weights. The fifth module was extraneous and tended to learn to imitate one of the other modules. After the MOSAIC model learned to produce the correct forces for the four training objects, we introduced the outlier, randomly interspersed among the training objects for 200 trials (the same number of trials as the test phase in our first experiment). Here, in the same format as Figure 2b, we see that the model has learned the outlier, unlike our participants. The force output for the outlier is much greater than the force expected by interpolating within the family. Moreover, examining the internal states of the individual model components shows that they are fully consistent with the explanation above: the module responsible for lifting the heaviest object rapidly comes to dominate the motor response to the outlier as well.

Although we could include this rather lengthy exposition of the MOSAIC model in our paper, we feel it would be distracting. Importantly, the main conclusion of this exercise is quite straightforward: the MOSAIC model effectively implements an associative map. Indeed, to our knowledge all current models of motor control predict associative map learning and do not account for categorical encoding. We recognize that this was not clear in our manuscript, especially for modular models like MOSAIC. In our revision, we have added a sentence to the Introduction clearly indicating that modular models implement associative map learning:

L99: “Note that associative map learning is also implemented by modular models, which have been proposed to account for learning in both motor ^30^ and non-motor ^31^ tasks.”

Additionally, we revisit this issue in the Discussion, where we have added the following text to point out that the recent developed COIN model also cannot, at present, explain our findings:

L694: “These various treatments can all be viewed as instances of non-parametric Bayesian models that leverage the hierarchical Dirichlet process, a statistically principled approach to clustering data into a theoretically infinite number of components ^125,126^. A recent motor control model has been developed based on this approach ^127^. However, at present this model cannot account for our results because there is no mechanism through which the visual properties of the objects—which are encoded as discrete cues—can be linked together to form a family.”

3. The basic model predictions are reasonably clear as is, but I agree with R3 that it isn't as clear as it could be whether or not block-level generalisation effects should exist in your data. Again, adding formal modelling would strongly address this ambiguity, but may not be necessary. If block level generalisation effects are predicted, then please report them in your revision.

We agree that the generalization analysis in our original paper was difficult to follow and unnecessarily complicated. In our original analysis, we compared the anticipatory force for the family objects in trials that came *immediately after* the outlier with a prediction of that force (using a linear model) based on other lifts of the family objects in the test phase. Moreover, we restricted our analysis to the two objects that were most visually similar to the outlier.

The reviewers suggested a far simpler “block-level” generalization analysis whereby we compare the anticipatory force produced for *all lifts* of the family objects during the test phase with the anticipatory force produced for the family objects at the end of the training phase. However, there is a problem with this approach. Any generalization from the outlier object to the family objects would be expected to be observed on the *first lift* of a family object following the lift of the outlier. However, once that family object has been lifted, any generalization from the outlier on additional lifts of family objects would be expected to be diminished. Indeed, in our response to Reviewer 3 below, we show this ‘wash-out’ of generalization in our data. Therefore, in our new generalization analysis, we focus on lifts of family objects that immediately followed the outlier, and compare the anticipatory force on these test phase lifts with corresponding lifts in the late training phase. We like this new approach because it is both sensible and straightforward. Note that the results of this new analysis are very clear: strong generalization is observed throughout the test phase in the Linear+ condition, whereas it is strongly diminished in the Linear++ condition, and absent in the Uncorr+ condition.

In the new analysis, we compare the anticipatory force on trials immediately following the test object with the average force from the final four bins of the training phase. Note that the trial order was controlled so that, every four trial cycles, each family object followed the outlier exactly once. As a result, the presentation frequency of the family objects is balanced, within each participant, for this analysis. Additionally, we have normalized our metric of generalization against the magnitude of the outlier (*i.e.*, we divide the observed increase in force by 2.94 N in the Linear+ and Uncorr+ groups, 5.89 N in the Linear++ group), in order to facilitate comparing the groups. We have modified Figure 2g-i and the accompanying text in the Results section to present this improved analysis:

L249: “The object families hypothesis predicts that when lifting an object that is encoded as a family member, the experienced density will update the density estimate for the family, thereby biasing the anticipatory force on a subsequent lift of a training (*i.e.*, family) object. […] For the Uncorr+ group (Figure 2i), we found no evidence of generalization at the start (*t*(11) = 1.90, p = 0.085) or the end of the test phase (*t*(11) = 0.45, p = 0.66), and no change over time (*t*(11) = 1.90, p = 0.084), consistent with encoding each object individually (Figure 2i).”

We have also replaced the previous description of the generalization analysis in the Methods with the following paragraph:

L909: “In the first experiment, we analyzed how lifting the test object generalized to the training objects in the subsequent trial (Figure 2g-i). […] For each group, we conducted two-tailed *t*-tests on this generalization metric in each portion of the test phase, and also on the change in this generalization metric from the early portion to the late portion of the test phase.”

4. R2 and R3 both raise the issue that there are useful reaction time predictions that should be investigated. In essence, we should see RT switch costs wherever a categorical effect is established and not elsewhere. In the revision, please test these predictions. If they are invalid, or the analysis isn't a good idea for some other reason, please explain why.

In our paper we reported that response times were greater in the Uncorr+ condition than in the Linear+ and Linear++ conditions combined. However, the source of this effect is not clear. The increased response time in the Uncorr+ condition could be a ‘switch’ cost related to loading a different category into motor memory. Alternatively, the increased response time could be a ‘selection’ cost associated with mapping the visual stimulus onto the appropriate category. Note that a selection cost would be expected to be considerably larger when lifting objects from five different categories (in the Uncorr+ condition) than when lifting objects from just two categories (in the Linear++ condition). However, a switch cost would not depend on the number of categories.

We can test between these explanations by examining the response time when lifting the outlier in the Linear++ condition. Note that, not surprisingly, we found that participants took longer to generate larger anticipatory forces in our task. To account for this uninteresting component of the response time, we fit a linear regression to response time as a function of object weight using data from the Linear+ and Linear++ conditions, and excluding trials with the test object and each immediately subsequent trial. This allowed us to determine the predicted response time, based solely on weight, which we then compared to the actual response time to test for additional temporal costs (e.g., associated with switching categories). We found a significant slope of 26.2 ms per Newton of anticipatory force. Therefore, when lifting the 1.5-kg test object in the Linear++ condition, we should expect the response time to be 154 ms longer than the average response time for the family objects (as the test object is 5.89 N heavier than the average family object weight). The actual response time for the 1.5-kg test object was 157 ms longer than the average response time for the family objects. In other words, the response time was 3 ms slower than the predicted response time based on weight. On trials that immediately followed the test object, which involved switching back to the memory of the family, the response time (averaging across objects) was 25 ms faster than the predicted response time. Neither of these differences was even close to being significant. As there was a category switch in motor memory in both of these cases, our data provide no evidence for the idea that the large temporal cost observed in the Uncorr+ condition is a switch cost. However, these results are consistent with a selection cost, which could be very large when dealing with five categories, but may be negligible in the trivial case involving only two categories.

We have included these additional response time analyses for the Linear++ group, along with our interpretation, in a new paragraph in the Results:

L286: “The increased response time in the Uncorr+ condition could be a ‘switch cost’ related to loading a different category in motor memory. […] In contrast, in the Linear++ condition, only two categories were involved, and therefore the selection cost may be negligible.”

We have also added the following text to the Methods, describing this response time analysis in detail:

L926: “In the Linear++ group, where we observed learning of separate categories, we analyzed whether response times were longer on trials involving a category switch (*i.e.*, trials with the test object or trials immediately after the test object). […] When analyzing trials immediately after the test object, we computed the predicted response time as the average response time for the family objects in ‘non-switch’ trials during the test phase, as the object weight in these trials was, on average, equivalent to the average family object weight.”

5. R2 and R3 both raise issues related to whether the lab-based task was bimodal. Please report this analysis in your revision, or else clearly outline why it isn't appropriate to do so (e.g., insufficient sample size etc.).

Thanks to a suggestion by Reviewer 2, we combined groups from the laboratory experiments to achieve sufficient sample sizes for this analysis. The results confirm that bimodal distributions of learners and non-learners were also observed in the laboratory experiments. We have included the following text after reporting the analysis for the web-based experiments:

L502: “Notably, similar bimodality was also observed in the laboratory experiments. Revisiting these data, we applied the same mixture model analysis to individual participants’ final outlier learning, (*i.e.*, the difference between the anticipatory force produced for the outlier and the family-predicted weight of the outlier). […] Thus, these results strengthen our main conclusion that memories of the motor properties of objects are organized categorically.”

We also added the following text to the Methods, after describing the analysis for the web-based task:

L1002: “The same analysis of bimodality was conducted for the laboratory experiments by combining the Linear+ and +Linear groups in one analysis, and the Linear++ and ++Linear groups in another analysis. For the laboratory experiments, we fit the single-Gaussian and two-Gaussian models to the distributions of the difference between anticipatory force for the outlier and the family-predicted weight of the outlier in the final 16 trial cycles of the test phase.”

Reviewer #1 (Recommendations for the authors):I have only two major recommendations for the authors.1. The paper is currently framed by asking "how memories of multiple objects might be organized", but I believe the paper would be clearer by using the following two passages from the discussion as anchors:"Current theories of motor learning often focus on graded generalization of learning across various stimulus and motor parameters as a revealing feature of the underlying computations 38,97-99. In particular, graded patterns of generalization have been taken as evidence that motor learning fundamentally involves associating contextual features of a movement with the target motor parameters in a continuous multi-dimensional space, often termed an associative map. The theoretical significance of our study is that it provides multiple, converging pieces of evidence for a fundamentally different type of organization-motor memories of objects are organized categorically, into families""These various treatments can all be viewed as instances of non-parametric Bayesian models that leverage the hierarchical Dirichlet process, a statistically principled approach to clustering data into a theoretically infinite number of components 123,124. Importantly, this approach has recently been applied to successfully account for an unprecedented range of phenomena in motor learning 125, suggesting that similar computations could also underlie the (in)ability to learn the weight of an outlier object in our lifting task."I think this framing would give you the opportunity explicitly walk the reader through the modelling of Wolpert and Kawato (1998) -- and any more recent motor learning theories that carry on in the internal model tradition and / or the context inference tradition -- such that we understand right from the beginning how the existence of categorical encoding would demand revision of our current thinking. If it turns out that the present data are completely consistent with current models, then you could use this framing to explain why the present data is nonetheless important. This would presumably entail a cogent explanation as to why the present data is itself novel. That is, I would like a clear description of the context effects studied in force fields and in any other relevant motor task such that the categorical effect you report can be seen to stand in sharp contrast.

We appreciate the idea of laying out relevant theoretical models at the beginning of a paper if they make opposing predictions, one of which is supported by the data. However, this is not how our findings relate to the two models discussed above. As explained in response to Essential Revision #2, the MOSAIC model is effectively an associative map model, and the existing Introduction addresses associative models and why they predict at least partial learning in all of our conditions. On the other hand, the relation to clustering models, such as the recently developed COIN model (Ref. 125 in the passage quoted above), is less straightforward. The COIN model is capable of generating new memories as needed; however, there is nothing in the model that links memories to categories and therefore the model, as it stands, cannot account for our data. Specifically, because sensory cues in the model are discrete labels, there is no notion of similarity between sensory cues (*e.g.*, objects of continuously varying height), and therefore no basis for forming families. It is possible that future extensions of the COIN model could explain the data, but that is not something we would like to speculate on in the current paper. Given these considerations, we feel strongly that the current approach to the Introduction, which gives a clear and simple intuition for the object families hypothesis and contrasts it with the associative map learning predicted by current models of motor control, is better than attempting to link the object families hypothesis to specific modeling concepts that, in any case, have not yet been sufficiently developed to account for our findings.

2. I believe the strength of the paper would be greatly improved if you performed modelling showing how and why the ideas of Wolpert and Kawato (1998) and related models are inadequate to capture the present data.Wolpert, D. M., and Kawato, M. (1998). Multiple paired forward and inverse models for motor control. Neural Networks, 11(7-8), 1317-1329.

As explained in response to Essential Revision #2, despite being modular, the MOSAIC model is essentially an associative map model that does not implement categorical encoding. Although it is possible that the model could be extended to accommodate categorical encoding, it is not clear to us how this could be done in a principled way. We feel it is most appropriate to leave this to future modeling work that carefully arbitrates between various implementations of categorical encoding. We do, however, explicitly point out in our revised manuscript that current models of motor control, including MOSAIC, predict associative map learning, and cannot account for categorical effects.

Reviewer #2 (Recommendations for the authors):I would like to raise several issues.1. In figure 2a (Linear+ group), the data of the 1st test trial (31st cycle) for the test object located in the just middle of the four training objects (i.e., ~9N). This result was reasonable from the idea of family effect: The motor system might estimate the weight by the interpolation. However, in figure 2c (Linear++ group), the participants failed to estimate the weight for the test object from the 1st test trial. This result was puzzling. Is there any reasonable explanation? Was there any possibility that this failure triggered to develop the different motor memory?

The Linear+ and Linear++ conditions were identical up to the first test trial, so there is no explanation for the apparent difference on the first test trial other than between-group differences due to random sampling. Indeed, the first test trial in Linear+ does not significantly differ from that in Linear++, nor does the first test trial in Linear++ significantly differ from the family-predicted weight. Thus, although the first test trial in the Linear++ condition does not appear to perfectly interpolate between the family members, our data are in fact consistent with interpolation.

2. Line 280-282 "These results suggest that….": If the processing objects in a family contribute to reducing the response time, I expect that the response time for the test object in Linear++ group should also exhibit a longer response time. Did the results support this prediction?

This is a reasonable prediction that we have now tested. Our analysis of the response times on ‘switch trials’ in the Linear++ group provided no evidence for increased response times (after accounting for the greater weight of the test object). As explained in detail in our response to Essential Revision #4, this analysis shows that the temporal cost observed in the Uncorr+ condition is not simply due to the process of loading a different category in motor memory. Instead, we argue that it is related to the process of mapping the visual stimulus onto the appropriate category (termed a ‘selection cost’). In the Uncorr+ group, participants were required to map each visual stimulus onto one of five categories in memory, and thus we would expect a substantial selection cost. In contrast, in the Linear++ condition, only two categories were involved, and therefore the selection cost may be negligible.

3. Line 390- "All-or-nothing learning of outlier weight": The results of the web-based task clearly indicated that the estimated weight distribution for Linear++ or Linear-- group was bimodal. Was bimodality also observed in the laboratory task? The number of participants for the Linear++ and ++Linear groups was sufficient to examine if the distribution was bimodal.

Thank you for calling this to our attention. We had overlooked the possibility of combining groups from the laboratory experiments to enable this analysis. As noted in our response to Essential Revision #5, we now include the results of this analysis, confirming that bimodal distributions of learners and non-learners were also observed in the laboratory experiments.

4. The web-based task: This is an innovative way of collecting data from many subjects. However, I wondered if this web-based task can be regarded as a motor learning task. In this task, the participants could directly see the motor output by the spring length. This was totally different from the laboratory experiment in which the participants could only use their haptic sensation. Considering that the participants had only to associate the object size with the spring length, the web-based task was like a test for declarative memory. If the web-based task was not a motor task, the object family effect could not be a property of motor memory but a property of declarative memory.

As explained in our response to Essential Revision #1, both tasks examine how individuals translate sensory information into continuous motor commands to achieve an action goal. Although the precise nature of the sensory information used for control differs between the tasks, both tasks engage very similar control mechanisms. We therefore feel strongly that the web-based task can be regarded as a motor task. We have added text to the Results and Discussion that explains the fundamental similarity between the tasks, and also points out that the use of visual feedback to set the initial conditions of a motor interaction is not uncommon in natural motor tasks such as archery, golf, and pinball.

Additionally, as we also explained in our response to Essential Revision #1, in our view, any memory, declarative or otherwise, used in the context of a motor task, can be viewed as a ‘motor memory’. In our revised manuscript, we now discuss the possible contributions of explicit and implicit processing in the encoding and use of categorical knowledge of object motor properties.

5. I was curious about how the participants recognized the error when lifting the test object. The difference in error awareness between Linear++ and Linear+ groups contributed to either creating a new memory or consistently using the same memory. I would also like to know if the participants noticed gradual weight change in Linear_up and Linear_down groups (Figure 4).

As explained in our response to Essential Revision #1, we did not ask participants to provide their explicit beliefs about each object’s weight during the course of the experiment, as this could have interfered with the processes we aimed to examine. As a result, there is little we can say regarding explicit awareness of errors and weights. We suspect that participants are aware of their error when lifting the test object even in the Linear+ condition, given that a 3-Newton error is very large, but do not attribute the error to the object. In general, how errors are credited likely involves a trade-off between acceptable errors and the cost of maintaining multiple categories in memory. In the revised Discussion, we comment briefly on the roles of implicit and explicit processes in weight prediction and memory.

Reviewer #3 (Recommendations for the authors):In this paper, Cesanek et al., use a novel object lifting task to investigate the "format" of memories for object dynamics. Namely, they ask if those memories are organized according to a smooth, local map, or discrete categories ('families'). They pit these competing models against one another across several experiments, asking if subjects' predicted weights of objects follow the family model or a smooth map. This was tested by having people train on objects of varying volumes/masses that were either consistent with a linear mapping between volume/mass, or where those dimensions were uncorrelated. This training phase was either preceded by, or preceded, a testing phase where a novel object with a deviant mass (but a medium-size volume) was introduced. As the authors expected, individuals trained on the linear mappings treated novel objects that were relatively close to the "family" average mass as a member of the family, and thus obligatorily interpolated to compute the expected mass of that object (i.e., under-predicting its true mass); conversely, when a novel object's mass was a substantial outlier w/r/t the training items, it was treated as a singleton and thus lifted with close to the correct force. Additional variations of this experiment provided further evidence that people tend to treat an object's dynamical features as a category label, rather than simply forming local associative representations. These findings offer a novel perspective on how people learn and remember the dynamics of objects in the world.Overall, I found this study to be both rigorous and creative. The experimental logic is refreshingly clear, and the results, which are replicated and extended several times in follow-up experiments, are rather convincing. I do think some additional analyses could be done, and data presentation could be improved. I also thought the generalization analysis, as I interpreted it, was difficult to align with the initial predictions.I think sequential effects in the test phase could be analyzed more closely. This is especially pertinent as the generalization metric appears to be based primarily on sequential effects (if I'm interpreting it correctly). Moreover, wouldn't the family model predict a block-level generalization effect? That is, if the family representation is a discrete memory, and is altered by incorporating the new, heavier object, wouldn't all items be influenced at a block-level, not just items nearby in volume/time? I think overall the connection between the family model and the particular implementation of the generalization analysis didn't fully click for me.

Thank you for this comment. We thoroughly examined the sequential effects in the test phase and arrived at a considerably improved generalization analysis, as explained in our response to Essential Revision #3. Please see that response for full details on the updated analysis, and the reasoning behind them.

This comment correctly points out that any generalization should affect all members of the family, not just the ones nearby in volume to the outlier. In the new generalization analysis, we now include all four family objects. However, our new analysis focuses on the lifts of the family objects that immediately follow the test object, and not the other lifts of the family objects (*i.e.*, lifts that occurred 2-4 lifts after the lift of the test object). We adopted this approach because generalization effects from the test object would be expected to diminish, thereby weakening our analysis. To verify this assumption, we analyzed the shape of the wash-out function in our experiment and found that although there are lingering effects of generalization in the second, third, and fourth trials, they are negligible compared to the first trial, and therefore it makes little sense to include them in the analysis (see ).

**Author response image 2. sa2fig2:** Generalization in each of the four trials after lifting the test object. The depicted mean +/- SEM for Trial #1 is the same as shown in Figure 2g-i. This figure demonstrates how the generalization effect rapidly washes out in the second, third, and fourth lifts after the test object, and thus why it is not desirable to include these trials in the generalization analysis.

I had trouble figuring out why there are different predictions for the family/association models in Figure 1F?

The pictorial differences in Figure 1F are intended only to reflect the qualitative distinction in how the two models encode the five objects, but the predictions for the two models are the same. We now briefly note this in the figure caption, and we have added some clarifying information to the relevant paragraph in the manuscript:

L88: “Under the associative map hypothesis, there is no fundamental difference between this scenario and those depicted in (d, e), so the prediction for learning is similar to the object families hypothesis (*i.e.*, all five objects will eventually be learned).”

L111: “Under the object families hypothesis, each of these objects is learned as an individual (Figure 1f; separate green lines) and, as a consequence, the training objects will be learned more slowly than when they are learned as a family and there will be minimal single-trial generalization from the ‘outlier’ to the training objects.”

Shouldn't response times for the Linear+/Linear++ conditions be different when subjects are faced with the novel object? (i.e., a memory "switch" cost?). This is hard to see here as they have been collapsed. Moreover, it could be useful to also analyze response times from stimulus appearance to initial force application (planning?).

This is a reasonable prediction, however our analysis of the response times on ‘switch trials’ in the Linear++ group provided no evidence for increased response times (after accounting for the greater weight of the test object). As explained in our response to Essential Revision #4, this analysis shows that the temporal cost observed in the Uncorr+ condition is not simply due to the process of loading a different category in motor memory. Instead, we argue that it is related to the process of selecting the appropriate memory from multiple candidates. This account explains the large effect in the Uncorr+ condition as well as the absence of a detectable effect in the Linear++ condition based on the notion that the selection process is trivial and rapid when there are only two categories (Linear++) but becomes time-consuming when there are five categories (Uncorr+).

Please note that participants’ force application prior to the button press was not constrained in any way and, as a result, it is impractical to attempt to measure reaction times as suggested.

The consistent undershoot in the novel object's mass for the Linear++/++Linear groups is treated, retrospectively, as the result of individual differences rather than a "true" undershoot (i.e., via the online experiment). Could this be confirmed in the original in-lab samples? I understand they are smaller Ns; at least it could be nice to see the individual-level data if the model fit is underpowered. (Indeed, individuals could be added to several of the summary statistic figures).

Based on a suggestion from Reviewer 2, we combined the Linear++ group with the ++Linear group and the Linear+ group with the +Linear group to achieve sufficient sample sizes to fit the mixture models. The results confirm that bimodal distributions of learners and non-learners were also observed in the laboratory experiments. Please see our response to Essential Revision #5 for the relevant text that has been included in the revision.

It seems plausible that the computation of an anticipatory force could come from parallel learning processes (e.g., very slow local map-learning + heuristic category judgment) within a single observer. Couldn't this also lead to the observed undershoot in the Linear ++ condition? While not critical, I think this point could be discussed somewhere.

In the revised Discussion, we now state that explicit processing is likely engaged when learning to re-classify the outlier object as an individual. In the same section, we point out that it is unclear whether explicit knowledge of object weight, which is known to be at least partially relative, is sufficient to produce lifting forces, which are absolute. Thus, while we do not discuss the specific possibility mentioned here (slow local map-learning + heuristic category judgment), we feel that this new Discussion paragraph addresses this comment by acknowledging potentially separate contributions of explicit and implicit processes in this task (*i.e.*, that explicit memory is likely engaged in object re-classification while implicit memory may be needed to map from category labels to absolute forces).